# Near-Exponential Savings for Mean Estimation with Active Learning

**Julian M. Morimoto**[*]
jmmorimoto@berkeley.edu

**Jacob Goldin**[†]
jsgoldin@uchicago.edu

**Daniel E. Ho**[‡]
dho@law.stanford.edu

## Abstract

We study the problem of efficiently estimating the mean of a $k$-class random variable, $Y$, using a limited number of labels, $N$, in settings where the analyst has access to auxiliary information (i.e.: covariates) $X$ that may be informative about $Y$. We propose an active learning algorithm ("PartiBandits") to estimate $\mathbb{E}[Y]$. The algorithm yields an estimate, $\widehat{\mu}_{\mathrm{PB}}$, such that $(\widehat{\mu}_{\mathrm{PB}} - \mathbb{E}[Y])^2$ is $\tilde{\mathcal{O}}\left(\frac{\nu + \exp(c \cdot (-N/\log(N)))}{N}\right)$, where $c > 0$ is a constant and $\nu$ is the risk of the Bayes-optimal classifier. PartiBandits is essentially a two-stage algorithm. In the first stage, it learns a partition of the unlabeled data that shrinks the average conditional variance of $Y$. In the second stage it uses a UCB-style subroutine ("WarmStart-UCB") to request labels from each stratum round-by-round. Both the main algorithm's and the subroutine's convergence rates are minimax optimal in classical settings. PartiBandits bridges the UCB and disagreement-based approaches to active learning despite these two approaches being designed to tackle very different tasks. We illustrate our methods through simulation using nationwide electronic health records. Our methods can be implemented using the **PartiBandits** package in R.

## 1 Introduction

Estimating the mean of a $k$-class random variable, $Y$, with limited data from a subset of the population of interest is a pervasive problem in statistics and machine learning. A classical solution to this problem is to draw a simple random sample (SRS) of $N$ independent and identically distributed (IID) labels and compute the resulting sample mean. However, this may be an inefficient use of the label budget if one has information $X$ (i.e., covariates) that may be related to $Y$. In such cases, one approach is to leverage $X$ to get a better estimate of $\mathbb{E}[Y]$ with fewer labels, perhaps through stratified random sampling (StRS) over $X$ and allocating the label budget across strata in proportion to how frequently each stratum occurs in the population. But in practice, there are many ways to define strata, and choosing a poor definition can result in minimal gains, or even worse performance than SRS. In general, analysts who use $X$ poorly, through stratification or otherwise, may over-sample some subpopulations and neglect others, resulting in biased or sub-optimally noisy estimates (*see, e.g.,* Aznag et al. (2023); Henderson et al. (2022)). This challenge has motivated the development of different adaptive sampling techniques for mean estimation (see, e.g., Seber and Mohammad Salehi (2015); Thompson (1991)), but these approaches focus on asymptotic performance and do not address whether fast rates of convergence can be achieved in finite samples. In parallel, the active learning literature has developed strategies for learning with limited labels. While classical active learning

---

[*]Department of Statistics, University of California, Berkeley; Regulation, Evaluation, and Governance Lab, Stanford Law School; World Bank Group

[†]University of Chicago; American Bar Foundation

[‡]Stanford University

39th Conference on Neural Information Processing Systems (NeurIPS 2025).

results primarily focus on classification (*see, e.g.,* Puchkin and Zhivotovskiy (2022); Hanneke and Yang (2014); Hanneke (2011)), recent work uses active learning to efficiently estimate subgroup (i.e. within-strata) means in settings where strata are predefined (Aznag et al., 2023). However, there has been no thorough exploration of active learning methods for population mean estimation when the researcher does not know an optimal stratification scheme. In this paper, we carry out such an exploration by developing an active learning framework for population mean estimation of $k$-class random variables, its convergence guarantees, and to what extent fast rates of convergence are achievable.

Our main problem setup revolves around estimating the mean of a $k$-class random variable $Y$, where the analyst has access to auxiliary information $X$ that may be informative about $Y$. The analyst may adaptively choose which instances to query for their corresponding labels, $Y$, round by round, with a budget of $N$ label requests. This setup parallels the pool-based active learning setup, where the analyst observes a large collection of IID unlabeled instances $X_1, X_2, \ldots$ and sequentially selects which ones to label, ultimately giving the analyst the labeled dataset, $(X_1, Y_1), \ldots, (X_N, Y_N)$. The hope is to obtain an estimate of $\mathbb{E}[Y]$ that is closer to $\mathbb{E}[Y]$ than the SRS strategies with high probability, where the latter convergence rates are on the order of $\mathcal{O}\left(\frac{\text{Var}(Y)}{N}\right)$.

There are two important reasons why it is hard to efficiently estimate population means in this problem setup. An ideal strategy would first partition the data into strata that minimize the average within-stratum variance of $Y$, then allocate the label budget across these strata according to the Neyman allocation to minimize the variance of the mean estimate that aggregates the subgroup mean estimates (*see* Jo et al. (2025); Bosch et al. (2003)). But in most settings, this optimal stratification is not known ahead of time. Moreover, variance within each stratum is not observed directly and must be estimated from noisy samples, so an allocation strategy that may seem optimal early on—based on preliminary variance estimates—may prove suboptimal as more data is collected. Thus, the analyst must (1) learn a good stratification from unlabeled data, and (2) decide how to allocate labels across strata adaptively in a way that reflects estimated (as opposed to oracle) variances.

## 1.1 Summary of Contributions

Our contributions are five-fold. First, we develop an active learning algorithm ("PartiBandits") for efficiently estimating the mean of a $k$-class variable $Y$. This algorithm yields an estimate, $\widehat{\mu}_{\text{PB}}$, such that $(\widehat{\mu}_{\text{PB}} - \mathbb{E}[Y])^2$ is $\tilde{\mathcal{O}}\left(\frac{\nu + \exp(c \cdot (-N/\log(N)))}{N}\right)$, where $c > 0$ is a constant and $\nu$ is the risk of the Bayes-optimal classifier (Theorem 3, Figure 1). It performs at least as well as SRS in $N$, and almost exponentially better when $X$ is predictive of $Y$ (i.e., when $\nu$ is small). It also closely resembles the exponential savings observed in disagreement-based active learning for classification (Puchkin and Zhivotovskiy, 2022; Hanneke and Yang, 2014; Hanneke, 2011), even though such results do not help with the task of mean estimation of $Y$ when $X$ does not perfectly predict $Y$ (*see* Dong et al. (2025)). Second, we show that if $X$ can be stratified in advance using a stratification scheme $\mathcal{G}$, the PartiBandits subroutine ("WarmStart-UCB") achieves error $\tilde{\mathcal{O}}\left(\frac{\Sigma_1(\mathcal{G})}{N}\right)$, where $\Sigma_1(\mathcal{G})$ is the average within-group variance of $Y$ (Theorem 1, Figure 1). Third, we show that both convergence rates are minimax optimal in classical settings (Theorems 2 and 4). Fourth, we bridge a gap between Upper Confidence Bound (UCB) algorithms and disagreement-based approaches in the active learning literature despite these two approaches being developed for very different tasks (Section 4.2). Finally, we conduct simulation studies using real-world data from over 6 million electronic health records and find that the gains predicted by our theory for population mean estimation can be achieved even in realistic small-sample regimes (Section 5).

## 2 Related Work

Our work builds on, and bridges, two different strands of prior work in active learning. The first is disagreement-based theory, as developed and refined by Hanneke (2011). This theory was originally designed for classification where labeled data are costly but unlabeled data are abundant. In this setting, the analyst queries labels for instances drawn from a large pool, concentrating effort on regions of the input space where candidate hypotheses disagree. A defining feature of this approach is its potential for "exponential savings", which refers to the convergence rate of excess classification error

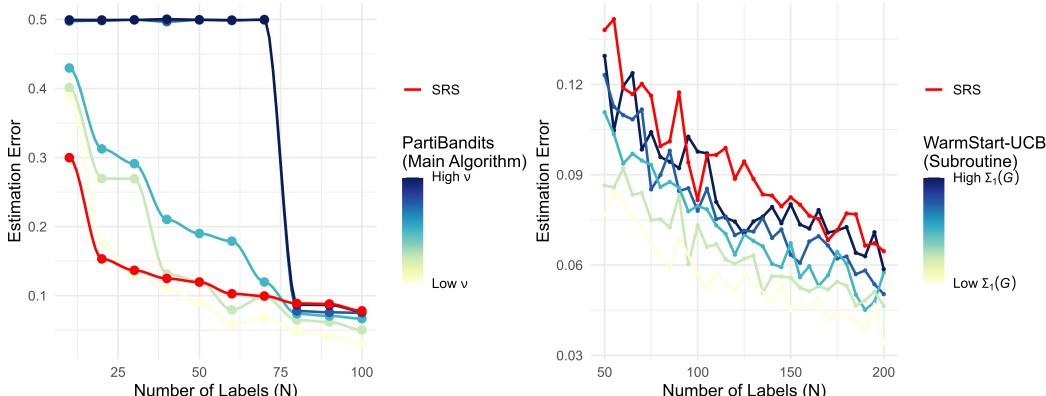

**Figure 1:** This plot compares the performance of PartiBandits and WarmStart-UCB, to SRS in different problem settings. The left panel compares SRS to PartiBandits for label budgets from 10 to 100. Here, $X \sim \text{Unif}[0, 1]$ and $Y = \mathbf{1}\{X \geq 0.5\}$, with a fixed fraction of $Y$'s (between 0% and 10%) randomly flipped to introduce noise. The proportion of flipped labels is equal to $\nu$ by definition. For each label budget, we generate 500 hypothetical datasets in this way, apply SRS and PartiBandits to each, and compute the resulting error rates. We then take the 90th percentile of these error rates to obtain a classical 90% high-probability/confidence bound. PartiBandits eventually outperforms SRS with relatively fewer samples, with performance gains becoming more pronounced when $X$ better predicts $Y$ and $\nu$ decreases. The right panel compares SRS to WarmStart-UCB for label budgets from 50 to 200. In this panel, $X \sim \text{Unif}[0, 1]$ and $Y = \mathbf{1}\{X \geq 0.5\}$, with 5% of the labels randomly flipped to introduce noise. We examine the effect of specifying different stratification schemes beforehand that reduce the within-group variance of $Y$ to varying degrees, where lower values of $\Sigma_1(\mathcal{G})$ indicate better average within-group variance reduction. Each scheme defines strata by applying a threshold between 0.3 and 0.5 and grouping observations based on whether $X$ falls to the left or right of the threshold. We run the same simulation procedure as above to obtain the 90% confidence bounds. WarmStart-UCB consistently outperforms SRS, and the gap grows when stratification reduces variance more effectively (i.e., when $\Sigma_1(\mathcal{G})$ shrinks).

(relative to the risk of the Bayes optimal classifier) shrinking at roughly $\tilde{\mathcal{O}}\left(\exp(c \cdot (-N/\log(N)))\right)$, far faster than the $\mathcal{O}(\text{Var}(Y)/N)$ rate typical in passive learning. While disagreement-based learning has been extensively studied in classification, it has not, to our knowledge, been applied to the problem of estimating population means. Our work is the first to show that the core insights of this framework can be used to construct stratification schemes that substantially reduce estimation variance in the mean estimation setting.

The second strand that our work connects to is UCB-style active learning. In particular, our proposed WarmStart-UCB subroutine is closely related to the work of Aznag et al. (2023), which developed a Variance-UCB algorithm for estimating the means of predefined subgroups using a fixed label budget, using upper confidence bounds on within-group variance to guide sampling. Our subroutine uses a similar approach to estimate the overall *population* mean using the strata selected by the first stage of our algorithm. Additionally, we build on the Aznag et al. (2023) results by showing in Theorem 1, that the rate of our subroutine for estimating population means from pre-defined strata explicitly quantifies the effect of how "informative" the subgroups are for estimating the quantity of interest, something that cannot be obtained through direct application of the main Aznag et al. results alone. Our rate also has improved dependence on key parameters such as the number of strata and $\sigma_{\min}$, the smallest conditional variance of $Y$ over all strata.

## 3 Notation and Problem Setups

### 3.1 Main Setup

Our main problem setup is that of estimating the population mean of a $k$-class random variable, $Y$, whose realizations come from the set $\{0, \ldots, k\}$ using a limited label budget $N$. Where appropriate, the realizations of $Y$ may also be any set with $k$ distinct elements in $\mathbb{R}$. The analyst has abundant access to unlabeled information $X \in \mathcal{X}$ (ex: covariates), which may be informative about $Y$, and can choose which examples to label in order to estimate $\mu := \mathbb{E}[Y]$. The analyst uses an algorithm to

construct an estimator $\widehat{\mu}(N)$ of the population mean $\mu$, that uses auxiliary information $X$ and only $N$ labels. The goal is to minimize the squared error, $(\widehat{\mu}(N) - \mu)^2$.

We also have the following terms and notations. A *hypothesis class* $\mathcal{C}$ is any set of measurable classifiers $h : \mathcal{X} \to \mathcal{R} \subset \mathbb{R}$ where $|\mathcal{R}|$ is finite. As we will show in Section 4.2, PartiBandits parallels classical disagreement-based active learning algorithms in that it requires $\mathcal{C}$ as an input. For any measurable $h : \mathcal{X} \to \{0, \ldots, k\}$, we define the squared loss of $h$ as $\mathrm{er}(h) = \mathbb{E}[(h(X) - Y)^2]$. Let $\nu = \inf_{h \in \mathcal{C}} \mathrm{er}(h)$, the *infimum loss* of $\mathcal{C}$.

As discussed in the Appendix, our main results still hold for other loss functions, including asymmetric misclassification costs. Such alternatives can produce more informative Bayes classifiers in applications where the ordinary squared-loss version fails to identify the threshold (such as when $\Pr(Y = 1 \mid X) \leq 1/2$ for all $X$).

Our main assumption is the following:

**Assumption 1** (Exponential Savings in Classification). *We assume that the joint of distribution $(X, Y)$ and the hypothesis class $\mathcal{C}$ are such that an active learning algorithm, $\mathcal{S}$, can be used to learn a classifier $\hat{h}$ such that with high probability, $\mathbb{E}[(\hat{h}(X) - Y)^2] - \nu \lesssim \exp\left(c \cdot \frac{-N}{\log(N)}\right)$, where $N$ is the label budget, and $c > 0$ is some $N$-independent constant.*

There are many problem setups in which this assumption is satisfied, as we discuss in Corollaries 1–4.

### 3.2 Setup for a PartiBandits Subroutine, WarmStart-UCB

Additionally, PartiBandits contains a subroutine that depends on the following problem setup that builds on the one discussed above. The following notation and definitions are drawn from Aznag et al. (2023). We assume that we can partition $\mathcal{X}$ using a stratification scheme $\mathcal{G} = \{A_1, \ldots, A_G\}$, where $A_g \subseteq \mathbb{R}^d$ are disjoint. Let $P_g = \mathbb{P}(X \in A_g)$ and $\mu_g = \mathbb{E}[Y \mid X \in A_g]\mathbb{P}(X \in A_g)$, so the population mean is $\mu = \sum_{g=1}^{G} \mu_g$. We define $\sigma_g^2 := \mathrm{Var}(P_g \cdot Y \mid X \in A_g)$, which equals $P_g^2 \cdot \sigma_g'^2$, where $\sigma_g'^2 := \mathrm{Var}(Y \mid X \in A_g)$ is the unweighted conditional variance of $Y$ given $X \in A_g$. The distribution of $X$ is assumed to be virtually known (as is the case in classical active learning setups), so $P_g$ is also known to the analyst. $\Sigma_1(\mathcal{G})$ is the average within-group variance of $Y$, $\sum_{g \in [G]} \sigma_g'^2 P_g$. The analyst wishes to compute an unbiased estimate of the population mean with $N$ label requests, sampling only one group from $\{A_1, \ldots, A_G\}$ at a time. The set of feasible policies for estimating $\mu$ is defined as $\Pi := \left\{\pi = \{\pi_t\}_{t \in [N]} \mid \pi_t \in G^{t-1} \times \mathbb{R}^{t-1} \to \Delta(G), \ \forall t \in [N]\right\}$, where $\Delta(G)$ is the set of measures supported on $[G]$. For some policy $\pi \in \Pi$, let $n_{g,N}(\pi)$ denote the number of collected samples from group $A_g$ after $N$ label requests by way of policy $\pi \in \Pi$, and let $\hat{\mu}_{g,N}(\pi)$ be the weighted sample mean estimator of $\mu_g$ for $n_{g,T}(\pi)$ collected samples, that is: $\widehat{\mu}_{g,N}(\pi) := \frac{1}{n_{g,N}(\pi)} \sum_{t : X_t \in A_g} Y_t \cdot P_g$. Once all data have been collected using the full label budget, $N$, the analyst will compute the population mean by aggregating the subgroup mean estimates obtained from the policy $\pi \in \Pi$: $\widehat{\mu}(\pi, N) = \sum_g \hat{\mu}_{g,N}(\pi)$. Going forward, we drop the explicit dependence on $\pi$ and $N$ in the notation when the policy is clear from context. As long as at least one sample is collected from each group, each $\widehat{\mu}_{g,N}$ is an unbiased estimator of $\mu_g = \mathbb{E}[Y \mid X \in A_g] \cdot P_g$, and hence the aggregated mean estimator $\widehat{\mu}$ is an unbiased estimator of the population mean $\mu$. The analyst wishes to obtain a high probability bound on the variance of $\widehat{\mu}$. The analyst does not know the true standard deviation vector $\boldsymbol{\sigma} := (\sigma_1, \ldots, \sigma_G)$, which can be used to obtain an upper bound on the variance of $\widehat{\mu}$ via $\mathrm{Var}(\widehat{\mu}) = \mathrm{Var}\left(\sum_{g=1}^{G} \widehat{\mu}_{g,N}\right) \leq \sum_{g=1}^{G} \frac{\sigma_g^2}{n_{g,N}}$. Therefore the analyst must learn $\boldsymbol{\sigma}$ through their decisions. We define the regret of a policy as $\mathrm{Regret}_N(\pi) := (\widehat{\mu}(\pi, N) - \mu)^2$.

## 4 Our Algorithms and Performance Guarantees

We now discuss our algorithms and their performance guarantees in turn. The proofs are in the Appendix. PartiBandits is our main algorithm, but since it incorporates a UCB-style subroutine, WarmStart-UCB, we first analyze the subroutine.

## 4.1 PartiBandits Subroutine: WarmStart-UCB

The first Algorithm is similar to the Variance-UCB algorithm of Aznag et al. (2023) except that we include an initial "warm-start" step (Step 1). We estimate $\sigma_g$ via the sample standard deviation, $\widehat{\sigma}_{g,t} := \sqrt{\frac{1}{n_{g,t}-1} \sum_{s \leq t : X_s = g} (P_g Y_s - \widehat{\mu}_{g,t})^2}$. We can then define $\mathrm{UCB}_t(\sigma_g) := \widehat{\sigma}_{g,t} + \frac{C_N(\delta)}{\sqrt{n_{g,t}}}$, where $C_N(\delta) := 2\sqrt{2c_1 \log\left(\frac{2}{\delta}\right) \log\left(\frac{c_2}{\delta}\right)} + \frac{2\sqrt{c_1 \log\left(\frac{2}{\delta}\right)(1 + c_2 + \log\left(\frac{c_2}{\delta}\right))}}{(1-\delta)\sqrt{2\log\left(\frac{2}{\delta}\right)}} \cdot \frac{1}{N^2}$. In $C_N(\delta)$, $c_1$ and $c_2$ are constant upper bounds on the sub-gaussian parameters of $Y$ (which exist since $Y$ is $k$-class), and $\delta \in (0,1)$ is parameter representing the confidence level for obtaining a high probability bound. WarmStart-UCB selects at each round the group with the largest upper confidence bound on its variance estimate, but begins with a "warm-start" phase that allocates a fixed fraction of the label budget, $\tau$, evenly across all groups (initiated by Step 1).

---

**Algorithm 1** WarmStart-UCB

---

**Require:** Label budget $N$, stratification scheme $\mathcal{G}$, confidence level $\delta$, buffer fraction $\tau$
1: Initialize $n_{g,0} = 0$, and $\hat{\sigma}_{g,t} = +\infty$ for all $g \in [G]$ and $t \leq \frac{\tau}{G} N$
2: Compute $C_N(\delta)$
3: **for** $t = 0, \ldots, N-1$ **do**
4:     Compute $\mathrm{UCB}_t(\sigma_g) = \hat{\sigma}_{g,t} + \frac{C_N}{\sqrt{n_{g,t}}}, \quad \forall g \in [G]$
5:     Select group $X_{t+1} = \arg\max_g \frac{\mathrm{UCB}_t(\sigma_g)}{n_{g,t}}$
6:     Observe feedback $Y_{t+1}$
7:     Update the number of samples: $n_{g,t+1} = n_{g,t} + \mathbf{1}_{X_{t+1}=g}, \quad \forall g \in [G]$
8:     Update the mean estimates, $\hat{\mu}_{g,t+1} = \frac{1}{n_{g,t+1}} \sum_{s=1}^{t+1} \mathbf{1}_{X_s=g} \cdot P_g Y_s, \quad \forall g \in [G]$
9:     Update the standard deviation estimates,
$$\hat{\sigma}_{g,t+1} = \sqrt{\frac{1}{n_{g,t+1}-1} \sum_{s \leq t+1 : X_s = g} (P_g Y_s - \hat{\mu}_{g,t+1})^2}, \quad \forall g \in [G]$$
10: **end for**
    **Output:** $\widehat{\mu}_{\mathrm{WS\text{-}UCB}}(N) = \sum_g \hat{\mu}_{g,N}.$

---

The following is an upper bound on the performance of WarmStart-UCB.

**Theorem 1.** $|\widehat{\mu}_{\textit{WS-UCB}} - \mathbb{E}[Y]|^2 = \tilde{\mathcal{O}}\left(\frac{\Sigma_1(\mathcal{G})}{N}\right).$

Theorem 1 shows that when a stratification scheme $\mathcal{G}$ is given *a priori*, WarmStart-UCB efficiently estimates the population mean with error scaling as $\tilde{\mathcal{O}}\left(\frac{\Sigma_1(\mathcal{G})}{N}\right)$, where $\Sigma_1(\mathcal{G})$ captures how informative the grouping is. The more informative the grouping, the smaller $\Sigma_1(\mathcal{G})$ is, and the faster the rate of convergence. By the law of total variance, $\Sigma_1(\mathcal{G}) \leq \mathrm{Var}(Y)$, so this rate is always at least as fast as that obtained with SRS. The proof is relatively straightforward. In Section A.5 of their work, Aznag et al. (2023) showed that $\frac{R_1(n) - R_1(n^*)}{R_1(n^*)} = \tilde{\mathcal{O}}(1/N)$. We do not define $R_1(n)$ and $R_1(n^*)$ explicitly here as this would involve significant technical detail. However, we show in the Appendix that $R_1(n)$ is equivalent to the variance of $\widehat{\mu}_{\mathrm{WS\text{-}UCB}}$, while $R_1(n^*)$ corresponds to $\Sigma_1(\mathcal{G})/N$. This identification allows us to directly leverage this bound from Section A.5 of Aznag et al. (2023). We then calculate how large $N$ must be in order for this quotient to be bounded from above by some constant (though this threshold is quite large, as it is inversely proportional to $\sigma_{\min}$), and we get a bound on the variance of $\widehat{\mu}_1$ for sufficiently large $N$. For all other $N$, we use the fact that a minimum fraction of the label budget is allocated to each group and obtain a similar high probability bound using classical Hoeffding arguments. This is why $\tau$ and the WarmStart step are important, as they safeguard the Variance-UCB procedure by ensuring that part of the label budget is allocated to StRS (every group gets a minimum number of samples), and this allows for nice convergence guarantees even when the proper rate of Aznag et al. (2023) does not hold. This allows us to obtain an analogous rate for label budgets that do not meet the threshold, thereby eliminating the counterintuitive dependence on $\sigma_{\min}$ that is typical in the active learning literature (Aznag et al., 2023; Carpentier et al., 2015). This ensures that our rate holds uniformly over all label budgets and constitutes a proper non-asymptotic, high-probability bound.

We note in the Appendix that when the dependence on the constants $\tau$ and $G$ is made explicit, the rate is $|\widehat{\mu}_{\text{WS-UCB}} - \mathbb{E}[Y]|^2 = \tilde{\mathcal{O}}\left(\frac{G \cdot \Sigma_1(\mathcal{G})}{N \cdot \tau}\right)$; however, we follow Aznag et al. (2023) in treating $G$ as a constant, and do the same for $\tau$. As we allude to above and show in the Appendix, the dependence on $\tau$ vanishes for large $N$ relative to $\sigma_{\min}$, and for all other $N$, the rate still holds with slightly larger constants (including a constant factor of $\tau^{-1}$). We also discuss in the Appendix the special cases when $\tau \in \{0, 1\}$.

While Aznag et al. (2023) established a rate of $\mathcal{O}(\tilde{N}^{-2})$ for the task of multi-group mean estimation (which can be extended to the task of population mean estimation) for a particular regret definition, the upper bound for WarmStart-UCB both (1) explicitly accounts for the signal of $Y$ in $X$ through $\Sigma_1(\mathcal{G})$, and (2) exhibits tighter dependence on the number of groups, $G$, and eliminates the dependence on the smallest conditional variance of $Y$ across all subgroups, $\sigma_{\min}$. The latter result in particular addresses an open problem in the active learning literature on mean estimation by demonstrating that not all active learning mean estimation frameworks result in the counterintuitive inverse dependence on $\sigma_{\min}$ (see, *e.g.*, Aznag et al. (2023); Carpentier et al. (2015); Ganti and Gray (2013)). While these improvements to the rate of convergence come at the cost of slower dependence on the label budget $N$ (from $N^{-2}$ to $N^{-1}$), this is expected as the $\mathcal{O}(\tilde{N}^{-2})$ of Aznag et al. (2023) is for a different definition of regret than the one we are interested in here, $(\widehat{\mu} - \mu)^2$.

The following provides a matching lower bound.

**Theorem 2** (Lower Bound for WarmStart-UCB). *Let $X \sim Unif[0, 1]$ and $Y = \mathbf{1}\{X \geq t\}$ for some $t \in [0, 1]$. Assume that a $\rho_\leq$-fraction of labels of $Y$ is flipped at random over $X \leq t$, and analogously with $\rho_>$ for $X > t$ and that $\rho_\leq, \rho_> < 1/4$. The stratification scheme $\mathcal{G}$ partitions the covariate space at the true threshold (i.e., groups $X < t$ and $X \geq t$). Then,*

$$|\widehat{\mu}' - \mathbb{E}[Y]|^2 \geq c_1 \frac{\Sigma_1(\mathcal{G})}{N}$$

*for some constant $c_1 > 0$ and all estimators $\widehat{\mu}'$ of $\mathbb{E}[Y]$ in this setup.*

Since this lower bound matches the upper bound of Theorem 1, we have that the rate of Theorem 1 is minimax optimal in this classical setting. This lower bound is based on the classical threshold example where the stratification scheme is such that the strata are chosen according to the decision boundary, and represents a favorable case where $X$ is highly predictive of $Y$ and the analyst has knowledge of how to group observations to reduce within-stratum variance—exactly the kind of setting any analyst would hope to operate in. The main point to note about this lower bound is that when the stratification scheme is well-chosen, the dependence in $N$ is still on the order of $1/N$. This will be important in the discussion of the lower bound for the main PartiBandits algorithm (Theorem 4).

### 4.2 PartiBandits

We now present our main algorithm, PartiBandits (Algorithm 2).

---
**Algorithm 2** PartiBandits

**Require:** hypothesis class $\mathcal{C}$, active learning classification algorithm $\mathcal{S}$, label budget $N$, confidence level $\delta$, buffer fraction $\tau$.
1: **Stage 1: Learn stratification using $\mathcal{S}$**
2: Run $\mathcal{S}$ with hypothesis class $\mathcal{C}$, label budget $\lfloor N/2 \rfloor$, and confidence level $\delta$ to obtain classifier $\widehat{h}$
3: Construct a stratification scheme $\mathcal{G}$ by defining $A_i = \widehat{h}^{-1}(i)$ for all $i \in \text{Im}(\widehat{h})$ and setting $\mathcal{G} = \{A_i\}_i$

4: **Stage 2: Apply Stratified Sampling Subroutine (WarmStart-UCB) to estimate means over $\mathcal{G}$**
5: Run WarmStart-UCB with label budget $N - \lfloor N/2 \rfloor$, stratification scheme $\mathcal{G}$ and buffer fraction $\tau$
   **Output:** $\widehat{\mu}_{\text{PB}} = \sum_g \widehat{\mu}_{g,N}$.

---

It is essentially a two-stage algorithm. In the first stage, it runs a disagreement-based algorithm, $\mathcal{S}$, that the analyst chooses. $\mathcal{S}$ helps identify a partition of the unlabeled data that shrinks the average

conditional variance of $Y$. In the second stage, it runs the WarmStart-UCB subroutine on that learned stratification. Examples of $\mathcal{S}$ to handle the case when $Y$ is binary ($k = 1$) include the $A^2$ algorithm of Balcan et al. (2006) and Algorithm 1 of Puchkin and Zhivotovskiy (2022). For the multiclass setting ($k > 1$), one may instead use algorithms such as Algorithm 1 of Agarwal (2013) to learn a partition of the unlabeled data reduces the mean conditional variance of $Y$. In Algorithm 2 we present PartiBandits with a general choice of $\mathcal{S}$, and show in Theorem 3 that it can achieve near-exponential savings whenever Assumption 1 is satisfied given the data-generating process, hypothesis class, and the choice of $\mathcal{S}$. We then illustrate in Corollaries 1-4 how different choices of S can accommodate different data-generating processes (e.g., binary vs. multiclass $Y$) and assumptions about the problem setup (such as the hard margin condition or the assumption that the Bayes optimal classifier is in the hypothesis class). The main theorem and its corollaries show that PartiBandits allows efficient mean estimation for multiclass outcomes across a wide range of structural assumptions and problem settings.

The following is an upper bound on the performance of Algorithm 2.

**Theorem 3.** *For any joint distribution of $(X, Y)$, hypothesis class $\mathcal{C}$, and $\mathcal{S}$ such that Assumption 1 holds, we have*

$$|\widehat{\mu}_{PB} - \mathbb{E}[Y]|^2 = \tilde{\mathcal{O}}\left(\frac{\nu + \exp(c \cdot (-N/\log(N)))}{N}\right),$$

*where $c > 0$ is a constant.*

We note in the Appendix that when the dependence on $\tau$ and $\mathcal{G}$ is made explicit, the rate is $|\widehat{\mu}_{\text{PB}} - \mathbb{E}[Y]|^2 = \tilde{\mathcal{O}}\left(|\mathcal{G}| \cdot \left(\frac{\nu + \exp(c \cdot (-N/\log(N)))}{N \cdot \tau}\right)\right)$ where $|\mathcal{G}|$ is the number of strata. Theorem 3 shows that PartiBandits efficiently estimates $\mathbb{E}[Y]$ by learning a stratification scheme $\mathcal{G}$ of $Y$ that yields an average within-stratum variance, $\Sigma_1(\mathcal{G})$, that is bounded from above by $\nu + \exp(c \cdot (-N/\log(N)))$. Asymptotically, this rate is faster than—or at least as fast as—the $1/N$ decay achieved by classical adaptive sampling methods (Félix-Medina, 2003; Thompson, 1991), since our bound decays at the rate of roughly $\exp(-cN/\log N)/N$ when $\nu$ is small.

*Result Intuition.* Disagreement-based active learning algorithms effectively learn a stratification scheme where within-group variance is reduced, since the labels in each stratum (i.e., strata induced by the inverse mapping of the classifier's prediction function) will tend to concentrate around a single class. Furthermore, they can do this with very few labels. Because active learning algorithms can identify these low-variance strata rapidly, we can then perform an adaptive stratified sampling procedure (Algorithm 1) using the learned stratification to estimate the population mean. Since estimation error depends primarily on the average within-group variance, reducing that variance quickly leads to a correspondingly fast decline in estimation error. We discuss in Corollary 4 how PartiBandits can further decompose relatively homogenous strata into sub-strata with higher and lower conditional variances, which allows the algorithm to allocate more samples to more heterogeneous sub-strata, yielding even better estimates of the population mean.

*Proof Sketch.* If Assumption 1 is satisfied, then there is an active learning algorithm, $\mathcal{S}$, such that when $\mathcal{S}$ is used in Step 2 of PartiBandits, we obtain a classifier, $\widehat{h}$, whose excess risk decays at an exponential rate, $\mathbb{E}[(\widehat{h}(X) - Y)^2] - \nu \lesssim \exp(c \cdot (-N/\log N))$ for some constant $c > 0$. We can then show that the variance of the mean estimate, $\widehat{\mu}_{\text{PB}}$, is bounded from above by $\mathbb{E}[(\widehat{h}(X) - Y)^2]$. In particular, we first use the law of total expectation to show that $\mathbb{E}[(\widehat{h}(X) - Y)^2] = \sum_{j \in J} \mathbb{E}[(j - Y)^2 \mid \widehat{h}(X) = j] \Pr(\widehat{h}(X) = j)$, where $J$ is the image of $\widehat{h}$. Then we use the Bias-Variance decomposition to show that the latter quantity is equal to $\sum_{j \in J} (\text{Var}(Y \mid \widehat{h} = j) + (j - \mathbb{E}[Y \mid \widehat{h} = j])^2) \Pr(\widehat{h} = j)$. Then it easily follows that this quantity is an upper bound on the average within-group variance of $Y$ using the stratification induced by $\widehat{h}$, and therefore an upper bound on the variance (and therefore the estimation error) of $\widehat{\mu}_{\text{PB}}$.

The constant $c$ depends on $\mathcal{C}$'s VC-dimension and disagreement coefficient (Hanneke, 2011). PartiBandits yields better mean estimates with smaller label budgets if $\mathcal{C}$ is well constructed and relatively small (as is the case when the analyst has good prior knowledge about possible ways in which $X$ may be related to $Y$), since less of the label budget is needed to eliminate incorrect hypotheses. It is typical for disagreement-based active learning algorithms to exhibit this dependence on the hypothesis class $\mathcal{C}$ (Puchkin and Zhivotovskiy, 2022; Hanneke and Yang, 2014).

With different choices of $\mathcal{S}$, we can obtain the following corollaries that allow for different assumptions and problem setups regarding the joint distribution of $(X, Y)$ and the hypothesis class $\mathcal{C}$. All proofs may be found in the Appendix.

**Corollary 1** (Classical Binary case with low noise). *Suppose $Y$ is binary, and the joint distribution of $(X, Y)$ and hypothesis class $\mathcal{C}$ are such that there exists $\mu < \infty$ such that for all $\varepsilon > 0$, $\mathrm{diam}(\varepsilon; \mathcal{C}) \leq \mu\varepsilon$, where $\mathrm{diam}(\varepsilon; \mathcal{C})$ is the diameter of the $\varepsilon$-minimal set of $\mathcal{C}$ (this is the "hard margin" condition. For further details, see Section 2 and Theorem 4 of Hanneke (2011)). If we set $\mathcal{S}$ to be the $A^2$ algorithm of Balcan et al. (2006), then, given a label budget of $N$ and $\delta \in (0, 1/2)$, we have with probability at least $1 - \delta$ that $|\widehat{\mu}_{PB} - \mathbb{E}[Y]|^2 = \tilde{\mathcal{O}}\left(\frac{\nu + \exp(c \cdot (-N/\log N))}{N}\right)$.*

**Corollary 2** (Binary, weaker structural conditions on $\mathcal{C}$). *Assume $Y \in \{0, 1\}$ and that the joint distribution of $(X, Y)$ and hypothesis class $\mathcal{C}$ are such that Massart's noise condition is satisfied (Assumption 4 in Puchkin and Zhivotovskiy (2022)), without requiring that $\mathcal{C}$ contain the Bayes optimal classifier. Suppose further that the joint distribution and hypothesis class are such that the star number $s$ and the (combinatorial) diameter of $\mathcal{C}$ are finite (see Section 2 and Theorem 4.1 of Puchkin and Zhivotovskiy (2022)). If we set $\mathcal{S}$ to be Algorithm 4.2 of Puchkin and Zhivotovskiy (2022), then, given a label budget of $N$ and $\delta \in (0, 1/2)$, we have with probability at least $1 - \delta$ that $|\widehat{\mu}_{PB} - \mathbb{E}[Y]|^2 = \tilde{\mathcal{O}}\left(\frac{\nu + \exp(c \cdot (-N/\log N))}{N}\right)$.*

**Corollary 3** (Multiclass). *Suppose $Y$ is $k$-class ($k > 2$) and that the joint distribution of $(X, Y)$ and hypothesis class $\mathcal{C}$ satisfy Assumptions 1–3 and the multiclass Tsybakov noise condition (Assumption 4) of Agarwal (2013). If we set $\mathcal{S}$ to be Algorithm 1 of Agarwal (2013), then, given a label budget of $N$ and $\delta \in (0, 1/e)$, we have with probability at least $1 - \delta$ that $|\widehat{\mu}_{PB} - \mathbb{E}[Y]|^2 = \tilde{\mathcal{O}}\left(\frac{\nu + \exp(c \cdot (-N/\log N))}{N}\right)$.*

Corollary 3 effectively allows PartiBandits to also handle real-valued outcomes (ex: $Y \sim \mathrm{Unif}[0, 1]$) if the analyst first discretizes $Y$ into bins, effectively turning the problem setup into that of Corollary 3.

Up to this point, we have focused on active learning algorithms $\mathcal{S}$ that guarantee exponential savings by grouping together instances that are likely to have similar labels. However, we may also consider $\mathcal{S}$ that not only identify homogeneous regions but also heterogeneous regions. Such $\mathcal{S}$ would be helpful for identifying strata for a distribution where labels are assigned by a simple threshold rule that outputs 0 if $x \leq 1/2$ and 1 otherwise, except that in the regions $x \in (1/4, 1/2]$ and $x \in (1/2, 3/4]$ the label is flipped with probability 0.1. The optimal stratification scheme here splits the domain into four intervals $[0, 1/4], (1/4, 1/2], (1/2, 3/4], (3/4, 1]$, and allocate more of the Stage-2 samples to the middle two strata where the labels are noisier. In Corollary 4 below, we introduce an example of an $\mathcal{S}$ that helps identify such a scheme. The proof is in the Appendix.

**Corollary 4** (Heterogeneity-Aware $\mathcal{S}$). *Assume the setup of Corollary 2. Define $\mathcal{S}$ in the following way:*

---

**Algorithm 3** Heterogeneity-Aware Active Learning Algorithm

---

**Require:** hypothesis class $\mathcal{C}$, label budget $N$, confidence level $\delta$.
 1: Run Algorithm 4.2 of Puchkin and Zhivotovskiy (2022) with a given label budget $N'$ and hypothesis class $\mathcal{C}$ to obtain a classifier, $\widehat{h}$
 2: Let $\mathcal{X}_*$ denote the abstention region obtained in Step 2 of Algorithm 4.2 of Puchkin and Zhivotovskiy (2022).
 3: Define $\widehat{h}^*(x) = \widehat{h}(x) + \varepsilon(N)$ if $x \in \mathcal{X}_*$, and $\widehat{h}^*(x) = \widehat{h}(x)$ otherwise. $\varepsilon(N)$ is an arbitrarily small number relative to $N$ (we may choose $\varepsilon(N) = \exp(-N/\log N)$).
 **Output:** $\widehat{h}^*(x)$

---

*Then we have that given a label budget of $N$ and $\delta \in (0, 1/2)$, we have with probability at least $1 - \delta$, $|\widehat{\mu}_{PB} - \mathbb{E}[Y]|^2 = \tilde{\mathcal{O}}\left(\frac{\nu + \exp(c \cdot (-N/\log N))}{N}\right)$.*

What this $\mathcal{S}$ does is capture heterogeneity via the abstention region $\mathcal{X}_*$ produced within Algorithm 4.2 of Puchkin and Zhivotovskiy (2022): the region where labels are more likely to be ambiguous. It

converts the final classifier $\widehat{h}$ returned by Algorithm 4.2 of Puchkin and Zhivotovskiy (2022) into a new classifier $\widehat{h}^*$ that takes values in $\{0, \varepsilon, 1, 1 + \varepsilon\}$, splitting the space into four strata that isolate both homogeneous and heterogeneous regions. The result is intuitive because the difference between $\widehat{h}$ from Corollary 2 and $\widehat{h}^*$ is small, so Assumption 1 is satisfied for $\widehat{h}^*$ if it is satisfied for that $\widehat{h}$.

The following yields a matching lower bound.

**Theorem 4** (Lower Bound for PartiBandits). *Consider the data-generating process where $X \sim Unif[0,1]$ and $Y = \mathbf{1}\{X \geq t\}$ for some $t \in [0,1]$, with a $\rho_{\leq}$- and $\rho_{>}$- fraction of labels $Y$ flipped at random over $X \leq t$ and $X > t$, respectively, and $\rho_{\leq}, \rho_{>} \leq 1/4$. Let $\mathcal{C} = \{\mathbf{1}\{(\cdot) \geq t\} : t \in [0,1]\}$. Then we have that for sufficiently large $N$,*

$$|\widehat{\mu} - \mathbb{E}[Y]|^2 \geq c_2 \frac{\nu + \exp(c \cdot (-N/\log(N)))}{N}$$

*for constants $c, c_2 > 0$ and all estimators $\widehat{\mu}$ of $\mathbb{E}[Y]$ in this setup.*

Since this lower bound matches the upper bound in Theorem 3, we have that the rate of Theorem 3 is minimax optimal for this classical setting. This lower bound is based on a simple threshold setup with segmented label noise. As we will show in Section 5, this setup reflects a common situation where a subset of the unlabeled data is highly predictive of $Y$. The proof of Theorem 4 starts by noting that the minimimum of $\Sigma_1(\mathcal{G})$ among all possible stratification schemes, $\mathcal{G}$, is precisely $\nu$ because of the way $Y$ is generated. We can then use the lower bounds in Hanneke and Yang (2014) and Hanneke (2011) to show that the exponential decay of the excess risk is the optimal rate in this setup, so no algorithm can cause $\Sigma_1(\mathcal{G})$ to converge to its optimal value faster than this exponential rate.

## 5 Empirical Illustration

We empirically evaluate the performance of our main algorithm, PartiBandits, and comparing it to SRS. We also test the WarmStart-UCB subroutine on the analogous mean estimation task when $X$ can be stratified according to some stratification scheme *a priori*. In most settings, SRS and stratified random sampling (StRS) are the standard approaches—and often the only realistic choices available—since the effectiveness of more sophisticated methods is highly domain- and application-specific. In practice, whether alternative sampling algorithms outperform SRS or StRS depends crucially on the relationship between observed covariates and the outcome of interest, which may not always be known or exploitable. That said, we present comparisons to other baselines, as well as analyses with other data generating processes, in the Appendix. We use Monte Carlo simulations and simulations involving nationwide electronic health records, with further details in the Appendix.

### 5.1 Simulations for Theorems 1 and 3

The error of the PartiBandits mean estimate is $\tilde{\mathcal{O}}\left(\frac{\nu + \exp(c \cdot (-N/\log(N)))}{N}\right)$, as shown in Theorem 3. Hence the critical parameter that affects this rate is $\nu$, which is closely related to $X$'s relationship with $Y$. The left panel of Figure 1 shows how PartiBandits performs as the strength of the relationship between $X$ and $Y$ varies. We see that generally, PartiBandits eventually outperforms SRS with relatively fewer samples, and this performance gap increases as the relationship between $X$ and $Y$ strengthens. We set $\mathcal{S} = A^2$ for our runs of PartiBandits. The right panel shows the performance of the WarmStart-UCB subroutine, which estimates the mean of $Y$ when $X$ can be stratified in advance according to some stratification scheme $\mathcal{G}$. We see that WarmStart-UCB consistently outperforms SRS when the stratification scheme closely aligns with the underlying decision boundary, effectively grouping observations with similar values of $Y$. We do not compare PartiBandits to $A^2$. $A^2$ is an active learning algorithm for classification, not mean estimation, so comparing its output directly to mean estimates from PartiBandits or SRS might not be meaningful, and attempting to do so can yield biased results Dong et al. (2025).

### 5.2 Simulations for Theorem 3 Using Health Records Data

To illustrate the gains of PartiBandits in a real-world setting, we leverage access to the American Family Cohort (AFC) dataset, which contains patient-level data from over 1,000 practices participating

in the American Board of Family Medicines PRIME Registry. AFC contains fine-grained longitudinal records of patient race and clinical diagnoses.

Our outcome of interest is a binary random variable, $Y = HB$, where $H$ indicates the presence of hypertension and $B$ is an indicator for whether the patient is Black. The estimand is $\mathbb{E}[Y]$, which corresponds to the fraction of patients who (1) are Black and (2) have hypertension. $X$ is the probability that an individual is Black based on their zip code, $Z$, which is available in the data. This setup reflects a common scenario in which researchers are studying demographic prevalences but lack access to direct demographic labels (Andrus et al., 2021; U.S. E.O., 2021), which motivates the use of proxy information like geolocation to guide sampling decisions. Though $X$ is defined as a probability mapping, it can also be viewed as an upper bound on the probability that $Y = 1$ for an individual from a certain zip code since $\Pr(HB = 1 \mid Z) \leq (\Pr(H = 1|Z) \wedge \Pr(B = 1 \mid Z))$, and therefore also a bound on the variance of $Y$. If $H$ is known but $B$ is not (analogous to the problem setting of Elzayn et al. (2025)), we can obtain even more sampling efficiency gains by forcibly setting $X = 0$ for all $H = 0$, since the variance of $Y$ is 0 for all such $H$. As a result, PartiBandits will not request labels $B$ from individuals such that $H = 0$. We compare the PartiBandits and SRS estimates of $\mathbb{E}[Y]$ under different label budgets. Because obtaining diagnosis and race data is often costly, this setting is such that labels are expensive, making it well suited to illustrate the benefits of using PartiBandits.

We focus on individuals whose derived probabilities of being Black, $X$, fall in the top and bottom 5th percentiles of the distribution. Although we restrict to these tails, this experimental setup does not simplify the problem. We are interested in the interaction variable $HB$ rather than a single class label, which makes identifying any separation more difficult. Even when $B = 1$, not all such individuals have hypertension, so $HB$ is not always 1 and can even be 0 more often than when $B = 0$. Since, AFC data are not IID draws from the general population, the upper tail may, for example, include individuals from predominantly Black but affluent neighborhoods who are less likely to have hypertension. Thus, $HB$ may even be 0 more often than for those classified as $B = 0$, so the class separation is not immediate. This setup reflects the reality of many datasets that are not IID samples from the general population.

Our focus on the tails also shows that $X$ does not need to be highly predictive of $Y$ across the entire population. We show that the mean within these tails where $X$ is highly predictive can be estimated accurately with much fewer labels than SRS requires, freeing up the remainder of the label budget for regions where $X$ is less informative of $Y$. This experimental setup thus illustrates how PartiBandits still offers a way of efficiently estimating the population mean.

To run this simulation, we draw, for each label budget, 500 random subsets of 10,000 patients each from the full AFC dataset of 6 million patients. Within each subset, we restrict attention to individuals whose geocoding-derived probabilities of being Black, $X$, fall in the top or bottom 5th percentile, and estimate the mean of $Y$ for this subpopulation using both PartiBandits and SRS. We compute the 90th percentile of the resulting estimation errors to obtain a high-probability bound for each method. Our choice of $\mathcal{S}$ is the classical $A^2$ algorithm of Balcan et al. (2006) (thus putting us in the regime of Corollary 1). Figure 2 shows the results and confirms that PartiBandits eventually outperforms SRS with relatively fewer samples. For smaller label budgets, SRS fares better but only by a universal constant factor.

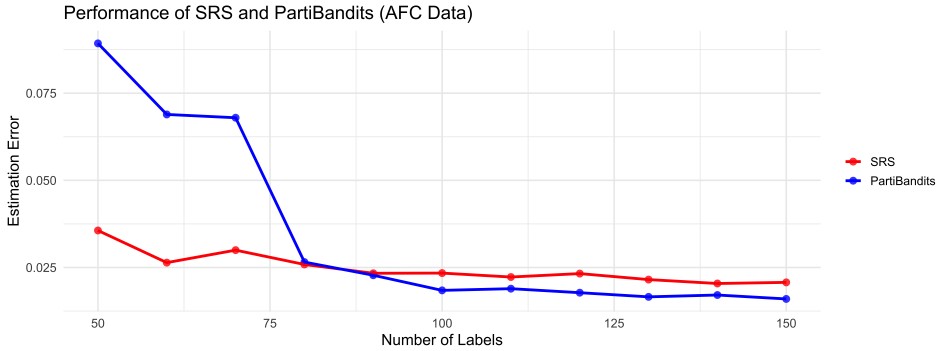

**Figure 2:** Comparison of estimation error for different label budgets using the AFC data.

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

# Appendix

## 5.3 Proofs

Before proving Theorem 1, we need a few auxiliary lemmas.

**Lemma 1.** *Let $\delta \in (0,1)$ and*

$$R_1(n_N) = R_1(n) := \left\| \left( \frac{\sigma_g^2}{n_{g,N}} \right)_{g=1}^{G} \right\|_{\ell_1},$$

*where $n_N = n = (n_{1,N}, \ldots, n_{G,N})$. Then we have that with probability at least $1 - \delta$,*

$$|\widehat{\mu}_{\textit{WS-UCB}} - \mu|^2 \leq C \left( R_1(n_N) \log \frac{2}{\delta} \right)$$

*for some absolute constant $C > 0$.*

*Proof.* We begin by recalling that for any policy we have:

$$\widehat{\mu}_{g,N} = \frac{1}{n_{g,N}} \sum_{t : X_t \in A_g} Y_t \cdot P_g,$$

and

$$\widehat{\mu}_{\text{WS-UCB}} = \sum_g \widehat{\mu}_{g,N}.$$

Since $\widehat{\mu}_{\text{WS-UCB}}$ is an unbiased estimator for $\mu$ and $\text{Var}(\widehat{\mu}_{g,N}) = \frac{1}{n_{g,N}^2} \sum_{t : X_t \in A_g} \text{Var}(Y_t P_g) = \frac{\sigma_g^2}{n_{g,N}}$, we have:

$$\Pr\left( |\widehat{\mu}_{\text{WS-UCB}} - \mu| \geq s \right) = \Pr\left( \left| \sum_g (\hat{\mu}_{g,N} - \mathbb{E}[\hat{\mu}_{g,N}]) \right| \geq s \right) \qquad \text{(Linearity)}$$

$$\leq 2 \exp\left( -\frac{s^2}{2 \sum_{i=1}^{g} \frac{\sigma_g^2}{n_{g,N}}} \right). \qquad \text{(Hoeffding)}$$

Through the classical exercise of setting the left-hand-side to $\delta$ and writing $s$ in terms of $\delta$, we have:

$$s^2 = 2 \underbrace{\sum_{g=1}^{G} \frac{\sigma_g^2}{n_{g,N}}}_{R_1(n_N)} \log \frac{2}{\delta}. \qquad (1)$$

$\square$

The next Lemma is simple but is important for linking the results of Aznag et al. (2023) to our work here.

**Lemma 2.** *Let*

$$R_1(n^*) := \frac{\left( \sum_{g \in [G]} \sigma_g \right)^2}{N}.$$

*Then,*

$$R_1(n^*) \leq G \frac{\Sigma_1(\mathcal{G})}{N},$$

*where $\Sigma_1(\mathcal{G})$ is the expected conditional variance of $Y$ given the stratification $\mathcal{G}$:*

$$\Sigma_1(\mathcal{G}) = \sum_{g \in [G]} \sigma_g'^2 P_g.$$

*Proof.* This follows immediately from the fact that $\sigma_g^2 = P_g^2 \cdot \sigma_g'^2$ and $P_g \in (0,1)$ for all $g \in [G]$ and from the norm equivalence property of $\ell^1$ and $\ell^2$ norms on $\mathbb{R}^d$. $\qquad\square$

*Proof of Theorem 1.* We have the following directly from Section A.5 of Aznag et al. (2023):

$$\frac{R_1(n) - R_1(n^*)}{R_1(n^*)} \leq \frac{G\|n - n^*\|_\infty^2}{N \min_g n_{g,N}^*} + \frac{7(3)^2 \Sigma_1^2}{\sigma_{\min}^2} \max_g \left(\frac{n_{g,N}^*}{n_{g,N}}\right)^6 \frac{\|n - n^*\|_\infty^3}{N^3},$$

where:

- $\sigma_{\min} = \min_{g \in [G]} \sigma_g$,

- $\Sigma_1 = \sum_{g \in [G]} \sigma_g$,

- $n_{g,N}^* = \frac{\sigma_g}{\Sigma_1} N$,

- and $n^* = \left(n_{1,N}^*, \ldots, n_{G,N}^*\right)$.

This simplifies to:

$$\frac{R_1(n) - R_1(n^*)}{R_1(n^*)} \leq \underbrace{\frac{G\|n - n^*\|_\infty^2}{N \min_g n_{g,N}^*}}_{(I)} + \underbrace{\frac{63\Sigma_1^2}{\sigma_{\min}^2} \max_g \left(\frac{n_{g,N}^*}{n_{g,N}}\right)^6 \frac{\|n - n^*\|_\infty^3}{N^3}}_{(II)}. \tag{2}$$

By Lemmas 1 and 2, it is sufficient to show that for sufficiently large $N$, the right hand side of Equation 2 is upper bounded by some constant. We first show that term $(I)$ is upper bounded by a constant and then we show that the same goes for term $(II)$. For all $N$ that are not sufficiently large, we will perform a classical Hoeffding analysis.

- **Bounding Term (I).** We have that

$$\|n - n^*\|_\infty \leq 3G + \frac{2GC_N}{\Sigma_1} \sqrt{\min_h n_{h,N}^*} \tag{3}$$

  by Equation 23 of Aznag et al. (2023), and

$$\min_h n_{h,N}^* = \frac{\sigma_{\min}}{\Sigma_1} N$$

  by the analysis in Lemma 1 of Aznag et al. (2023). What we want to do first is combine the summation on the right hand side of 3 into one term. As long as

$$N \geq \underbrace{\left(2\Sigma_1 \sqrt{\frac{\Sigma_1}{\sigma_{\min}}}\right)^2}_{C_1(\Sigma, \sigma_1)}, \tag{Condition 1}$$

then:

$$\sqrt{\min_h n^*_{h,N}} = \sqrt{\frac{\sigma_{\min}}{\Sigma_1} N}$$

$$\geq 2\Sigma_1, \qquad\qquad \text{(by Condition 1)}$$

which implies that:

$$\frac{2GC_N}{\Sigma_1} \sqrt{\min_h n^*_{h,N}} \geq 4G \geq 3G, \qquad\qquad (C_N \geq 1 \text{ by construction})$$

and so we can write:

$$\|n - n^*\|_\infty \leq \frac{8GC_N}{\Sigma_1} \sqrt{\min_h n^*_{h,N}}.$$

Thus, if we assume Condition 1 and

$$N \geq \underbrace{\frac{64G^3 C_N^2}{\Sigma_1^2}}_{C_2(G, \Sigma_1, \sigma_{\min})}, \qquad\qquad \text{(Condition 2)}$$

then we have that

$$\underbrace{\frac{G\|n - n^*\|_\infty^2}{N \min_g n^*_{g,N}}}_{(I)} \leq \frac{G\left(\frac{8GC_N}{\Sigma_1} \sqrt{\min_h n^*_{h,N}}\right)^2}{N \cdot \min_g n^*_{g,N}}$$

$$= \frac{G \cdot \left(\frac{64G^2 C_N^2}{\Sigma_1^2} \cdot \min_h n^*_{h,N}\right)}{N \cdot \min_g n^*_{g,N}}$$

$$= \frac{64G^3 C_N^2}{\Sigma_1^2 N}$$

$$\leq 1.$$

- **Bounding Term (II).** We rewrite this term as the product of two terms, $\left(\frac{63\Sigma_1^2}{\sigma_{\min}^2} \frac{\|n - n^*\|_\infty^3}{N^3}\right) \cdot \left(\max_g \left(\frac{n^*_{g,N}}{n_{g,N}}\right)^6\right)$, and focus on each term in the product separately.

We first consider the term $\frac{63\Sigma_1^2}{\sigma_{\min}^2} \frac{\|n - n^*\|_\infty^3}{N^3}$. We observe that if we assume Condition 1 and

$$N \geq \underbrace{\left(\frac{200^2 \cdot G^3 C_N^3}{\Sigma_1^{11/2} \sigma_{\min}^{1/2}}\right)^{2/3}}_{C_3(G, \Sigma_1, \sigma_{\min})}, \qquad\qquad \text{(Condition 3)}$$

then

$$\begin{aligned}
N^3 &\geq \frac{200^2 \cdot G^3 C_N^3}{\Sigma_1^{11/2} \sigma_{\min}^{1/2}} \cdot N^{3/2} && \text{(by Condition 3)} \\
&\geq \frac{32256 \cdot G^3 C_N^3}{\Sigma_1^{11/2} \sigma_{\min}^{1/2}} \cdot N^{3/2} \\
&= \frac{63 \cdot 512 \cdot G^3 C_N^3}{\Sigma_1^{11/2} \sigma_{\min}^{1/2}} \cdot N^{3/2} \\
&= \frac{63 \Sigma_1^2}{\sigma_{\min}^2} \cdot \frac{512 G^3 C_N^3}{\Sigma_1^3} \left( \frac{\sigma_{\min}}{\Sigma_1} N \right)^{3/2} \\
&= \frac{63 \Sigma_1^2}{\sigma_{\min}^2} \cdot \left( \frac{8 G C_N}{\Sigma_1} \sqrt{\frac{\sigma_{\min}}{\Sigma_1}} N \right)^3 \\
&\geq \frac{63 \Sigma_1^2}{\sigma_{\min}^2} \cdot \|n - n^*\|_\infty^3, && \text{(by Condition 1)}
\end{aligned}$$

which implies

$$\frac{63 \Sigma_1^2}{\sigma_{\min}^2} \frac{\|n - n^*\|_\infty^3}{N^3} \leq 1.$$

Next, we consider the term $\max_g \left( \frac{n_{g,N}^*}{n_{g,N}} \right)^6$. We have by the proof of Theorem 1 in Aznag et al. (2023) that if $N$ is sufficiently large such that $\frac{\|n - n^*\|_\infty}{\min_h n_{h,N}^*} \leq 1$, then

$$\max_g \frac{n_{g,N}^*}{n_{g,N}} \leq \frac{1}{1 - \frac{\|n - n^*\|_\infty}{\min_h n_{h,N}^*}}.$$

Hence, we have that as long as Condition 1 is satisfied and

$$N \geq \underbrace{\frac{(16 G C_N)^2}{\Sigma_1 \sigma_{\min}}}_{C_4(G, \Sigma_1, \sigma_{\min})}, \qquad \text{(Condition 4)}$$

then, dividing both sides of that inequality by $2\sqrt{N}$, we have

$$\begin{aligned}
\frac{1}{2} &\geq \frac{8 G C_N}{\sqrt{\Sigma_1 \sigma_{\min}} N} \\
&= \frac{\frac{8 G C_N}{\Sigma_1} \sqrt{\frac{\sigma_{\min}}{\Sigma_1}} N}{\frac{\sigma_{\min}}{\Sigma_1} N} \\
&= \frac{\|n - n^*\|_\infty}{\min_h n_{h,N}^*},
\end{aligned}$$

and therefore we have:

$$\max_g \left( \frac{n_{g,N}^*}{n_{g,N}} \right)^6 \leq \left( \frac{1}{1 - 1/2} \right)^6 \leq 64.$$

This gives us that:

$$\frac{63\Sigma_1^2}{\sigma_{\min}^2} \max_g \left(\frac{n_{g,N}^*}{n_{g,N}}\right)^6 \frac{\|n - n^*\|_\infty^3}{N^3} \le 1 \cdot 64$$

if Conditions 1, 3, and 4 hold.

Hence, if we let:

$$C(G, \Sigma_1, \sigma_{\min}) := \max\left\{C_1(\Sigma_1, \sigma_{\min}), C_2(G, \Sigma_1, \sigma_{\min}), C_3(G, \Sigma_1, \sigma_{\min}), C_4(G, \Sigma_1, \sigma_{\min})\right\}$$

then we have that as long as $N \ge C(G, \Sigma_1, \sigma_{\min})$, then

$$\frac{R_1(n) - R_1(n^*)}{R_1(n^*)} \le 1 + 64$$
$$\le 65,$$

which means that

$$R_1(n) \le C' \cdot R_1(n^*)$$

for some absolute constant $C' > 0$, and so:

$$(\widehat{\mu}_{\text{WS-UCB}} - \mu)^2 \le C' \left(R_1(n_N) \log \frac{2}{\delta}\right)$$
$$\le C' \cdot R_1(n^*) \cdot \log \frac{2}{\delta}$$
$$\le \left(C' \log \frac{2}{\delta}\right) G \frac{\Sigma_1(\mathcal{G})}{N}. \qquad \text{(Lemma 2)}$$

Now for the analysis when $N < C(G, \Sigma_1, \sigma_{\min})$. By construction of Algorithm 1, we have that for all $g \in [G]$:

$$n_{g,N} \ge \frac{\tau}{G} N.$$

So by 1 we have:

$$(\widehat{\mu}_{\text{WS-UCB}} - \mu)^2 \le 2 \sum_{g=1}^G \frac{\sigma_g^2}{n_{g,N}} \cdot \log \frac{2}{\delta} \qquad (4)$$
$$\le C \cdot \frac{G}{\tau} \sum_{g=1}^G \frac{\sigma_g^2}{N} \cdot \log \frac{2}{\delta} \qquad (5)$$
$$\le \left(C \cdot \frac{G}{\tau} \log \frac{2}{\delta}\right) \frac{\Sigma_1(\mathcal{G})}{N}. \qquad (6)$$

We follow Aznag et al. (2023) in treating $G$ as a constant, and do the same with $\tau$ to obtain:

$$(\widehat{\mu}_{\text{WS-UCB}} - \mu)^2 = \tilde{\mathcal{O}}\left(\frac{\Sigma_1(\mathcal{G})}{N}\right).$$

$\square$

**Remark 1** (Dependence on $\tau$ and $G$.). *We note that by 6, when the dependence on the constants $\tau$ and $G$ is made explicit, the rate is*

$$|\widehat{\mu}_{\textit{WS-UCB}} - \mathbb{E}[Y]|^2 = \tilde{\mathcal{O}}\left(\frac{G \cdot \Sigma_1(\mathcal{G})}{N \cdot \tau}\right).$$

*We discuss the dependencies on $\tau$ and $G$ further below.*

*Proof of Theorem 2.* The proof follows from the fact that the Neyman allocation corresponding to this $\mathcal{G}$ yields the smallest value of $\Sigma_1(\mathcal{G}')$ over all possible stratification schemes $\mathcal{G}'$ and classical results about the estimation of the mean of IID Bernoulli random variables (ex: Lemma 1 from Hanneke and Yang (2010)). The idea is that first, $\Sigma_1(\mathcal{G})$ minimizes $\Sigma_1(\mathcal{G}')$ among all possible stratification schemes $\mathcal{G}'$ (precisely because $\mathcal{G}$ aligns with the way the data are generated) and the fastest possible rate for estimating the conditional mean of $Y$ on each stratum of $\mathcal{G}$ is simply the conditional variance divided by the number of labels requested on that stratum (ex: Lemma 1 from Hanneke and Yang (2010)). Combining the results yields the bound. $\square$

*Proof of Theorem 3.* The proof essentially amounts to showing that the value of $\Sigma_1(\mathcal{G})$ produced by Algorithm 2 is bounded from above by $\nu + \exp(c \cdot (-N/\log(N)))$. To do this, we first show that:

$$\mathbb{E}[(\widehat{h}(X) - Y)^2] \geq C \cdot \Sigma_1(\mathcal{G}).$$

We first use the law of total expectation to obtain that:

$$\mathbb{E}[(\widehat{h}(X) - Y)^2] = \sum_{j \in J} \mathbb{E}[(j - Y)^2 \mid \widehat{h}(X) = j] \Pr(\widehat{h}(X) = j) \tag{7}$$

where $J$ is the image of $\widehat{h}$. We will now proceed with a bias-variance decomposition. We have that for all $j \in J$,

$$\mathbb{E}\big[(\widehat{h}(X) - Y)^2 \mid \widehat{h}(X) = j\big] = \mathbb{E}\big[(j - Y)^2 \mid \widehat{h}(X) = j\big].$$

Define

$$\mu_j := \mathbb{E}[Y \mid \widehat{h}(X) = j].$$

Then we have that

$$\mathbb{E}[(j - Y)^2 \mid \widehat{h}(X) = j] = \mathbb{E}[(j - \mu_j + \mu_j - Y)^2 \mid \widehat{h}(X) = j].$$

Expanding using the identity
$$(a - c + c - b)^2 = (a - c)^2 + (c - b)^2 + 2(a - c)(c - b),$$
we get:

$$\mathbb{E}[(j - \mu_j + \mu_j - Y)^2 \mid \widehat{h}(X) = j] = (j - \mu_j)^2 + \mathbb{E}[(\mu_j - Y)^2 \mid \widehat{h}(X) = j]$$
$$+ 2(j - \mu_j) \cdot \mathbb{E}[(\mu_j - Y) \mid \widehat{h}(X) = j].$$

Note that the last term, $2(j - \mu_j) \cdot \mathbb{E}[(\mu_j - Y) \mid \widehat{h}(X) = j]$, is 0 by linearity of expectation and the definition of $\mu_j$.

So ultimately we have that:

$$\mathbb{E}\big[(\widehat{h}(X) - Y)^2 \mid \widehat{h}(X) = j\big] = (j - \mu_j)^2 + \mathbb{E}[(\mu_j - Y)^2 \mid \widehat{h}(X) = j].$$

Now using 7 we can rewrite:

$$\mathbb{E}[(\widehat{h}(X) - Y)^2] = \sum_{j \in J} \left[ (j - \mu_j)^2 + \mathbb{E}[(\mu_j - Y)^2 \mid \widehat{h}(X) = j] \right] \cdot \Pr(\widehat{h}(X) = j). \tag{8}$$

If we let $\sigma_j'^2 = \mathrm{Var}(Y \mid A_j)$ for each $j \in J$, then by definition of $\mu_j$ we have that:

$$\mathbb{E}[(\widehat{h}(X) - Y)^2] \geq \sum_{j \in J} \sigma_j'^2 \Pr(\widehat{h}(X) = j)$$
$$= \Sigma_1(\mathcal{G}).$$

Now by Assumption 1, we have that:

$$\mathbb{E}[(\widehat{h}(X) - Y)^2] \leq \nu + \exp(c \cdot (-N/\log(N))),$$

so we have that:

$$\Sigma_1(\mathcal{G}) \leq \nu + \exp(c \cdot (-N/\log(N)))$$

for some constant $c > 0$. Applying Theorem 1 with our choice of $\mathcal{G}$ and the remaining label budget of $N/2$ we obtain that:

$$(\widehat{\mu}_{\mathrm{PB}} - \mathbb{E}[Y])^2 = \tilde{\mathcal{O}} \left( \frac{\nu + \exp(c \cdot (-N/\log(N)))}{N} \right).$$

$\square$

**Remark 2** (More on the Dependence on $\tau$ and $G$ ($|\mathcal{G}|$))**.** *We note that by Remark 1, it is straightforward to show that when the dependence on $\tau$ and $\mathcal{G}$ is made explicit, the rate is $|\widehat{\mu}_{PB} - \mathbb{E}[Y]|^2 = \tilde{\mathcal{O}} \left( |\mathcal{G}| \cdot \left( \frac{\nu + \exp(c \cdot (-N/\log(N)))}{N \cdot \tau} \right) \right)$ where $|\mathcal{G}|$ is the number of strata.*

**Remark 3** (Different Loss Functions)**.** *We note that Theorem 3 still holds if one is interested in an asymmetric misclassification cost, $a \cdot \mathbf{1}\{h(X) = 0, Y = 1\} + b \cdot \mathbf{1}\{h(X) = 1, Y = 0\}$ instead of the ordinary squared loss. We assume here that $Y$ is binary.*

*Proof.* To see why we first start by doing an analogous decomposition to 7:

$$\mathbb{E} \left[ a \cdot \mathbf{1}\{\widehat{h}(X) = 0, \ Y = 1\} + b \cdot \mathbf{1}\{\widehat{h}(X) = 1, \ Y = 0\} \right]$$
$$= \mathbb{E} \left[ b \cdot \mathbf{1}\{Y = 0\} \mid \widehat{h}(X) = 1 \right] \cdot P(\widehat{h}(X) = 1) + \mathbb{E} \left[ a \cdot \mathbf{1}\{Y = 1\} \mid \widehat{h}(X) = 0 \right] \cdot P(\widehat{h}(X) = 0),$$

and this is precisely equal to:

$$= b \cdot \mathbb{E} \left[ (\widehat{h}(X) - Y)^2 \mid \widehat{h}(X) = 1 \right] \cdot P(\widehat{h}(X) = 1) + a \cdot \mathbb{E} \left[ (\widehat{h}(X) - Y)^2 \mid \widehat{h}(X) = 0 \right] \cdot P(\widehat{h}(X) = 0).$$

Therefore the rest of the proof for this alternative misclassification cost is analogous to the proof of Theorem 3, just with adjustments for the constants $a$ and $b$. $\square$

This result is helpful in a setting where $X \sim \mathrm{Unif}[0, 1]$ and

$$Y \sim \begin{cases} \mathrm{Bern}(0), & X \leq 0.5, \\ \mathrm{Bern}(0.25), & X > 0.5. \end{cases}$$

Here, an alternative loss would be helpful because the typical Bayes optimal classifier based on the squared loss would not distinguish between $X \leq 0.5$ and $X \geq 0.5$, but the Bayes optimal classifier resulting from an asymmetric misclassification loss would.

*Proof of Corollary 1.* By Theorem 5 of Hanneke (2011), $\mathcal{S} = A^2$ learns a classifier with exponential savings in Stage 1 of PARTIBANDITS. Hence Assumption 1 is satisfied, and Theorem 3 follows. $\square$

*Proof of Corollary 2.* By Theorem 4.1 of Puchkin and Zhivotovskiy (2022), Algorithm 4.2 of Puchkin and Zhivotovskiy (2022) learns a classifier with exponential savings in Stage 1 of PARTIBANDITS. Hence Assumption 1 is satisfied, and Theorem 3 follows. $\square$

*Proof of Corollary 3.* As demonstrated by Corollary 1 of Agarwal (2013), Algorithm 1 of Agarwal (2013) learns a classifier with exponential savings in Stage 1 of PARTIBANDITS. Hence Assumption 1 is satisfied, and Theorem 3 follows. $\square$

*Proof of Corollary 4.* We assume without loss of generality that $\exp\left(\frac{-N}{\log(N)}\right) \leq 1$ (the result can be easily adjusted when this assumption does not hold).

By Theorem 3, it is sufficient to show that Assumption 1 is satisfied:

$$\mathbb{E}[(\hat{h}^*(X) - Y)^2] - \nu \lesssim \exp\left(c \cdot \frac{-N}{\log(N)}\right),$$

for some constant $c > 0$. For this, it is sufficient to show that the excess risk of $\widehat{h}$ and $\widehat{h}^*$ differ by at most $\exp\left(c \cdot \frac{-N}{\log(N)}\right)$, since $\widehat{h}$ already satisfies assumption 1 (Corollary 2). Formally, this means showing that:

$$\left|\mathbb{E}[(\hat{h}(X) - Y)^2] - \mathbb{E}[(\hat{h}^*(X) - Y)^2]\right| \lesssim \exp\left(c \cdot \frac{-N}{\log(N)}\right).$$

We have that:

$$
\begin{aligned}
\left|\mathbb{E}[(\hat{h}(X) - Y)^2] - \mathbb{E}[(\hat{h}^*(X) - Y)^2]\right| &= \left|\mathbb{E}\left[(\hat{h}(X) - Y)^2 - (\hat{h}^*(X) - Y)^2\right]\right| \quad \text{(linearity)} \\
&= \left|\mathbb{E}\left[\hat{h}(X)^2 - 2\hat{h}(X)Y + Y^2 - \hat{h}^*(X)^2 + 2\hat{h}^*(X)Y - Y^2\right]\right| \\
&= \left|\mathbb{E}\left[\hat{h}(X)^2 - \hat{h}^*(X)^2 - 2Y\left(\hat{h}(X) - \hat{h}^*(X)\right) : \widehat{h} \neq \widehat{h}^*\right]\right| \\
&\leq \left|\mathbb{E}\left[\hat{h}(X)^2 - \hat{h}^*(X)^2 : \widehat{h} \neq \widehat{h}^*\right]\right| \\
&\quad + 2\left|\mathbb{E}\left[Y\left(\hat{h}(X) - \hat{h}^*(X)\right) : \widehat{h} \neq \widehat{h}^*\right]\right|. \\
&\qquad\qquad\qquad\qquad\qquad\qquad\qquad \text{(triangle inequality)}
\end{aligned}
$$

We note that $2\left|\mathbb{E}\left[Y\left(\hat{h}(X) - \hat{h}^*(X)\right) : \widehat{h} \neq \widehat{h}^*\right]\right| \leq c_1 \exp\left(\frac{-N}{\log(N)}\right)$ for some $c_1 \geq 0$ by construction of $\widehat{h}^*$. Furthermore, we have using the difference of squares decomposition that:

$$
\begin{aligned}
\left|\mathbb{E}\left[\hat{h}(X)^2 - \hat{h}^*(X)^2 : \widehat{h} \neq \widehat{h}^*\right]\right| &= \left|\mathbb{E}\left[(\hat{h}(X) - \hat{h}^*(X))(\hat{h}(X) + \hat{h}^*(X)) : \widehat{h} \neq \widehat{h}^*\right]\right| \\
&\leq 2\mathbb{E}\left[\left|(\hat{h}(X) - \hat{h}^*(X))\right| : \widehat{h} \neq \widehat{h}^*\right],
\end{aligned}
$$

where the inequality follows from Jensen's inequality, definition of $\widehat{h}^*$, and the assumption that $\exp\left(\frac{-N}{\log(N)}\right) \leq 1$. Again by definition of $\widehat{h}^*$, we have that $2\mathbb{E}\left[\left|(\hat{h}(X) - \hat{h}^*(X))\right| : \widehat{h} \neq \widehat{h}^*\right] \leq c_2 \exp\left(\frac{-N}{\log(N)}\right)$ for some $c_2 > 0$. Thus, we have that $\left|\mathbb{E}\left[\hat{h}(X)^2 - \hat{h}^*(X)^2 : \widehat{h} \neq \widehat{h}^*\right]\right| \leq c_2 \exp\left(\frac{-N}{\log(N)}\right)$, and therefore that:

$$\left|\mathbb{E}[(\hat{h}(X) - Y)^2] - \mathbb{E}[(\hat{h}^*(X) - Y)^2]\right| \lesssim \exp\left(c \cdot \frac{-N}{\log(N)}\right).$$

$\square$

*Proof of Theorem 4.* As shown in Theorem 2, the best possible performance achievable by any algorithm is $\frac{C}{\sqrt{N}}$ for some constant $C > 0$ that may depend on other parameters of the problem. So determining the lower bound for any algorithm returns an estimate of $\mu$ becomes a task of determining the lower bound of $C > 0$ for any algorithm. We were able to deduce the lower bound of $C$ in Theorem 2 by restricting to a particular situation (that in which the analyst basically is able to leverage $X$ perfectly in a sense). We consider here a more general problem setting where the analyst is limited to estimators $\widehat{\mu}$ that compute weighted aggregates of stratum means over partitions consisting of exactly two strata, $S_0$ and $S_1$, which make up some stratification scheme $\mathcal{G}$. The analyst learns $S_0$ and $S_1$ through some procedure after $N/2$ label requests, and uses the rest of the label budget to estimate the means within each stratum and then aggregating them at the end. This is a best case scenario because this is aligns with the data generation process and helps approximate the optimal value of $C > 0$ (one only needs to learn the best choice of $S_0$ and $S_1$ to arrive at the optimal $C$, as shown in Theorem 2). Hence, the lower bound on $(\widehat{\mu} - \mathbb{E}[Y])^2)$ is simply the lower bound on $\Sigma_1(\mathcal{G})$, so we focus on the latter quantity in this proof.

Assume without loss of generality that $\mu_0 = \mathbb{E}[Y \mid X \in S_0] \leq 1/2$ and $\mu_1 = \mathbb{E}[Y \mid X \in S_1] \geq 1/2$. There exists a classifier $h$ such that $h^{-1}(0) = S_0$ and $h^{-1}(1) = S_1$. Recall from 8 that we have the following for any $h$:

$$\mathbb{E}[(h(X) - Y)^2] = \left((1 - \mu_1)^2 + \mathbb{E}[(\mu_1 - Y)^2 \mid h(X) = 1]\right) \cdot P(h(X) = 1)$$
$$+ \left(\mu_0^2 + \mathbb{E}[(\mu_0 - Y)^2 \mid h(X) = 0]\right) \cdot P(h(X) = 0)$$

Since $\mu_0 \leq 1/2$ and $\mu_1 \geq 1/2$, we have that $(1 - \mu_1)^2 \leq (1 - \mu_1)\mu_1$ and $\mu_0^2 \leq \mu_0(1 - \mu_0)$. Thus, we have that:

$$\mathbb{E}[(h(X) - Y)^2] \leq \left((1 - \mu_1)\mu_1 + \mathbb{E}[(\mu_1 - Y)^2 \mid h(X) = 1]\right) \cdot P(h(X) = 1)$$
$$+ \left(\mu_0(1 - \mu_0) + \mathbb{E}[(\mu_0 - Y)^2 \mid h(X) = 0]\right) \cdot P(h(X) = 0)$$

Since $Y$ is Bernoulli, we have by the corresponding variance formula that:

$$\mathbb{E}[(h(X) - Y)^2] \leq 2\mathbb{E}[(\mu_1 - Y)^2 \mid h(X) = 1] \cdot P(h(X) = 1)$$
$$+ 2\mathbb{E}[(\mu_0 - Y)^2 \mid h(X) = 0] \cdot P(h(X) = 0)$$
$$\leq 2\Sigma_1(\mathcal{G}).$$

What this shows is that if we can lower bound $\mathbb{E}[(h(X) - Y)^2]$, then we can lower bound $\Sigma_1(\mathcal{G})$. In their discussion of Theorem 4, Hanneke (2011) noted that the rate:

$$\mathbb{E}[(h(X) - Y)^2] - \nu \leq +\exp(c \cdot (-N/\log(N)))$$

is minimax optimal in precisely this situation where the strata map on to threshold classifiers (*see also* Castro and Nowak (2008); Cohn et al. (1994); Burnashev and Zigangirov (1974)), so we have that there is a $D \in \mathcal{D}$ such that with probability at least $\delta$,

$$\mathbb{E}_D[(h(X) - Y)^2] \geq K\left(\nu + \exp(c \cdot (-N/\log(N)))\right),$$

for some absolute constant $K > 0$, and therefore that:

$$2\Sigma_1(\mathcal{G}) \geq K\left(\nu + \exp(c \cdot (-N/\log(N)))\right),$$

for $(X, Y) \sim D$. $\square$

**Additional Details Regarding Simulations**

**Details Regarding Simulations for Figure 1**

In both simulations, each dataset contains 10,000 unlabeled instances. For each label budget from 80 to 140, we run 500 simulations. For each run, we compute the absolute difference between the mean estimated by the algorithm and the true mean; we then report the 90th percentile of these 500 errors.

**Details Regarding AFC Simulations**

In each simulation, we construct the mapping from ZIP code to the probability of being Black using the 10,000-patient sample drawn for that run. This means that the probability mapping $X$ is re-estimated in every simulation based on the ZIP code–race distribution within the sampled subset.

**Additional Notes**

**Note 1** (Notes about Theorem 1). *When the dependence on the constants $\tau$ and $G$ is made explicit, the rate is*

$$|\widehat{\mu}_{\textit{WS-UCB}} - \mathbb{E}[Y]|^2 = \tilde{\mathcal{O}}\left(\frac{G \cdot \Sigma_1(\mathcal{G})}{N \cdot \tau}\right).$$

*Following Aznag et al. (2023), we treat $G$ as a constant, and do the same for $\tau$. As discussed above and shown here, the dependence on $\tau$ vanishes for large $N$ relative to $\sigma_{\min}$, and for all other $N$, the rate still holds with slightly larger constants (including a linear factor of $\tau^{-1}$). We also discuss the special cases when $\tau \in \{0, 1\}$.*

*Setting $\tau = 0$ allocates all labels to Variance-UCB, achieving optimal rates when both $\sigma_{\min}$ and the label budget are sufficiently large. For sufficiently large $N$ relative to $\sigma_{\min}$, $\tau$ is no longer relevant to the problem and the ordinary rate holds thanks to the main result of Aznag et al. (2023). However, a positive $\tau$ ensures that at least some samples are drawn uniformly from every group, preserving the guarantee of Theorem 1 with only slightly larger constants. For large label budgets, the effect of $\tau$ disappears and the optimal rates of Aznag et al. (2023) apply. Since the rate's dependence on the label budget remains unchanged (only the constants differ), we present the simplified rate $\tilde{\mathcal{O}}\left(\frac{\Sigma_1(\mathcal{G})}{N}\right)$ in Theorem 1.*

*When $\tau$ is small (close to 0), we rely more heavily on the Variance-UCB adaptive sampling strategy, which is reasonable if the label budget is large and/or $\sigma_{\min}$ is not too small. When $\tau$ is large (close to 1), the procedure defaults to simple stratified random sampling, ensuring the stated dependence on the label budget in the theorem but resulting in slightly less optimal constants. In the final bounds, $\tau$ essentially appears as a constant in the denominator, but for sufficiently large label budgets, its effect becomes negligible as the guarantees of the Variance-UCB procedure dominate.*

*The WarmStart in Algorithm 1 serves as a buffer, ensuring that Variance-UCB remains effective when the overall label budget is small and the algorithm might otherwise undersample low-variance groups. As Aznag et al. (2023) discuss, the regret of Variance-UCB can scale inversely with $\sigma_{\min}$, the smallest group variance. By assigning each group a minimum number of samples, the warmstart helps guarantee the algorithm's fast-rate properties. For small $N$, the risk bound behaves as $\tilde{\mathcal{O}}(1/\tau N)$, where $\tau$ is the warmstart proportion. As $N$ grows and the fast-rate conditions hold, the dependence on $\tau$ vanishes and the sharper rates of Aznag et al. (2023) apply. We follow Aznag et al. (2023) in focusing only on the dependence on the label budget in the bound.*

**Note 2** (Note about Empirical Illustration.). *Here present additional experiments using alternative data-generating processes, varying degrees of class separation, and asymmetric class distributions. We also consider the comparison of PartiBandits to alternative baselines.*

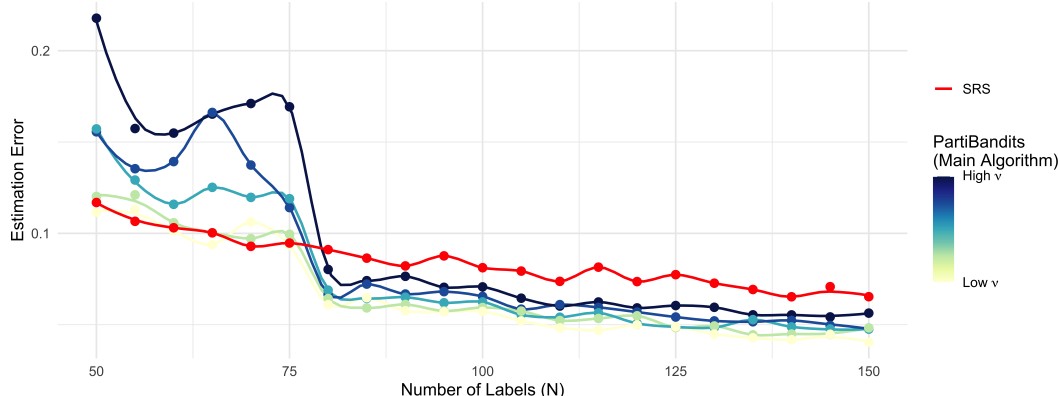

**Figure 3:** This plot compares the performance of PartiBandits to SRS when the labels are generated according to the following logistic data generating process: $X \sim \mathrm{Unif}[0,1]$ and $Y \sim \mathrm{Bernoulli}\left(\frac{1}{1+\exp[-(\beta_0+\beta_1 X)]}\right)$, where $\beta_0 = -1/\nu$ and $\beta_1 = 2/\nu$. This corresponds to a Logit-type DGP, with $1/\nu$ governing the steepness of the logistic curve. For each label budget, we generate 500 hypothetical datasets in this way, apply SRS and PartiBandits to each, and compute the resulting error rates. We then take the 90th percentile of these error rates to obtain a classical 90% high-probability/confidence bound. PartiBandits eventually outperforms SRS with relatively fewer samples, with performance gains becoming more pronounced when $X$ better predicts $Y$ and $\nu$ decreases.

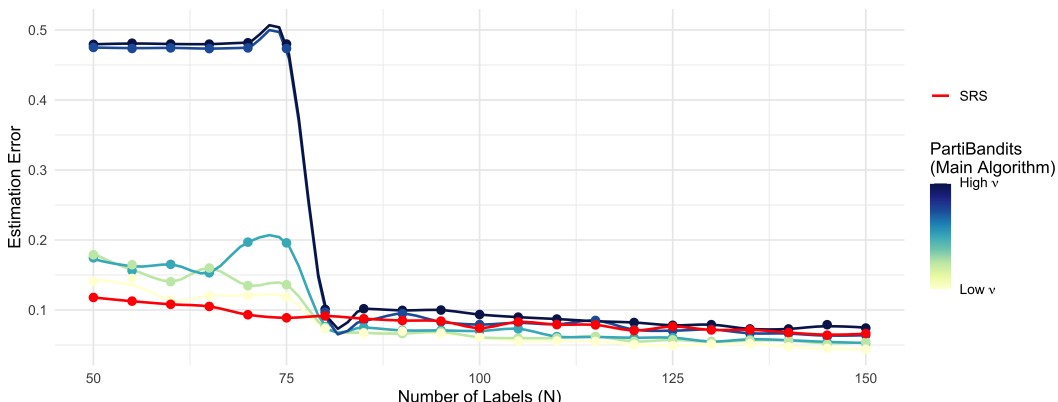

**Figure 4:** This plot compares the performance of PartiBandits to SRS when the labels are generated according to the following asymmetric probit data generating process: $X \sim \mathrm{Unif}[-5,5]$ and $Y \sim \mathrm{Bernoulli}\big(\Phi\big((1/\nu)\,(X - 0.25)\big)\big)$, where $\Phi(\cdot)$ denotes the standard normal CDF. This corresponds to a Probit-type DGP, with $1/\nu$ controlling the steepness of the probability curve and $X \approx 0.25$ marking the midpoint threshold. For each label budget, we generate 500 hypothetical datasets in this way, apply SRS and PartiBandits to each, and compute the resulting error rates. We then take the 90th percentile of these error rates to obtain a classical 90% high-probability/confidence bound. PartiBandits eventually outperforms SRS with relatively fewer samples, with performance gains becoming more pronounced when $X$ better predicts $Y$ and $\nu$ decreases.

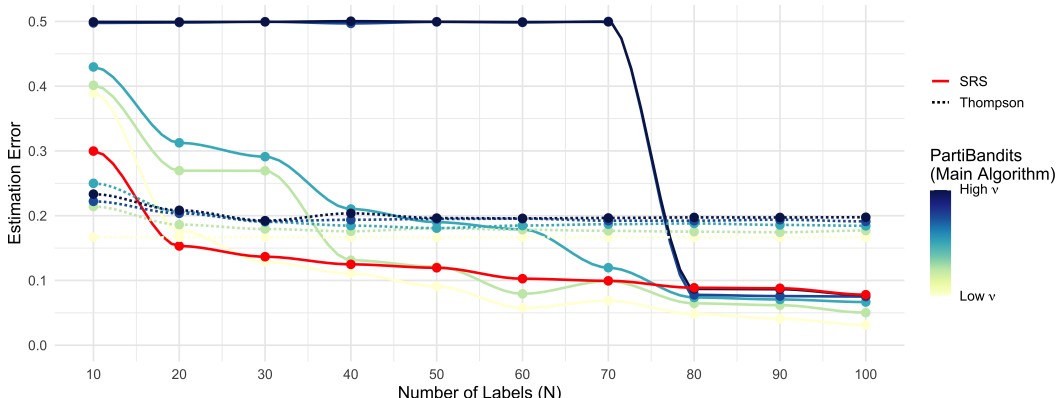

**Figure 5:** This plot compares the performance of PartiBandits to SRS and Thompson sampling and SRS for label budgets from 10 to 100. Here, $X \sim \text{Unif}[0, 1]$ and $Y = \mathbf{1}\{X \geq 0.5\}$, with a fixed fraction of $Y$'s (between 0% and 10%) randomly flipped to introduce noise. The proportion of flipped labels is equal to $\nu$ by definition. For each label budget, we generate 500 hypothetical datasets in this way, apply SRS, Thompson sampling, and PartiBandits to each, and compute the resulting error rates. To execute the Thompson sampling, we use the standard Beta-Bernoulli Thompson Sampling algorithm with an uninformative prior $\text{Beta}(1, 1)$. At each round, the algorithm samples a success probability from each arm's posterior, selects the arm with the highest draw, observes a Bernoulli reward, and updates the corresponding posterior. In our setup, we ran $T = 3000$ rounds with $K = 3$ arms (true $p = (0.1, 0.5, 0.8)$) for the prototype and $K = 5$ bins over $[0, 1]$ with a threshold of 0.5 for the binned variant. We then take the 90th percentile of these error rates to obtain a classical 90% high-probability/confidence bound. PartiBandits eventually outperforms SRS and Thompson sampling with relatively fewer samples, with performance gains becoming more pronounced when $X$ better predicts $Y$ and $\nu$ decreases. We also observe that, over time, Thompson sampling ceases to yield better mean estimates, consistent with theoretical results suggesting that this procedure can yield biased mean estimates in common settings (Shin et al., 2019).

