# OpenReview forum: "Near-Exponential Savings for Population Mean Estimation with Active Learning"
_NeurIPS.cc/2025/Conference — NeurIPS 2025 poster_

### Official Review · Reviewer_MZfC · 2025-06-15

**Clarity:** 2
**Significance:** 2
**Originality:** 3
**Rating:** 4
**Confidence:** 4

**Summary:**

This paper proposes PartiBandits, an active learning algorithm for estimating the population mean of a binary label with covariates and limited labels. The method combines stratified sampling with an adaptive allocation policy, and achieves a near-exponential convergence in Massart low-noise condition. Matching upper and lower bounds are provided and experiments on semi-synthetic data validate the gains under ideal stratification.

**Questions:**

What is the convergence rate if the assumption is not Massart low noise condition (i.e. hard-margin) but rather more general Tsybakov condtion?

The authors eliminate the dependence on \sigma_{min} but introduce another buffer \tau, what is the specific dependence on \tau in the final bounds? Is there a trade-off between big \tau and small \tau?

Why is A^2 not used in the evaluation, is there any practical concern, e.g. efficiency?

**Ethical Concerns:**

["NO or VERY MINOR ethics concerns only"]

**Final Justification:**

The authors managed to make a significant addition to more general cases on the top of hard margin condition and binary setup, explain the role of \tau more explicitly, add the Thompson sampling baseline, and revise the confusing notations. This is impressive and I tend to raise my score.

**Limitations:**

Yes.

**Paper Formatting Concerns:**

NA.

**Quality:**

2

**Strengths And Weaknesses:**

Strengths:

Tackles an under-explored but important problem: active learning for population mean estimation.

Both matching lower bounds and upper bounds are given, making the algorithms minimax optimal.

Given a pre-defined stratification G, WS-UCB can achieve nearly exponential convergence rate when \mu is small.

Empirical validation, though limited, shows clear improvements over SRS.

Weaknesses:

The main result only considers the hard-margin condition, which is rarely the case in practice.

It is not clear what is the role of buffer \tau in the final bound of theorem 3

The stratification process is under-specified: A^2 is mentioned in theory but not used in experiments.

Experiments have only SRS baseline.

Minor presentation issues: e.g. what are \rho_{\leq} and \rho_{>} in the statements of Theorem 4? Some notations, like R_1(n),  lack definition in the main text.

---

> ### Author Rebuttal · Authors · 2025-07-31
>
> Thank you for your thoughtful and detailed review. Your feedback highlights several important issues, and we have used your comments to guide our planned revisions and clarifications to the paper. Below, we address each of your points, with the \textbf{main responses highlighted in bold}. After reviewing these clarifications and planned updates, we hope you will consider whether a re-evaluation of the score would be appropriate.
>
> \subsection{Strengths and Weaknesses}
>
> \textbf{W1: Thank you for pointing this out. We recognize that the hard-margin condition is restrictive in practice. Accordingly, we have extended the main theorem to cover more general cases, and Corollary 2 provides an example where the hard-margin condition is not required. Please see Point 0 and point W2, and point Q3 of our response to Reviewer ZBCc for details.}
>
> \textbf{W2: Thank you for bringing this to our attention. Please see points W4 and Q3 in our response to Reviewer ehtb for our response and planned revisions in response to this feedback.}
>
> \textbf{W3}: \textbf{Thank you for your question about the stratification process and the use of $A^2$. You are correct—our experiments do not include a line corresponding solely to $A^2$, since we use it only as a subroutine within the PartiBandits algorithm, not as an independent baseline. $A^2$ is an active learning algorithm for classification, not mean estimation, so comparing its output directly to mean estimates from PartiBandits or SRS might not be meaningful (see lines 69–71 of the manuscript). We use $A^2$ to identify covariate regions likely to have homogeneous outcomes, which then define strata for adaptive sampling in mean estimation. Although one could construct a mean estimate directly from the classifier’s output, this approach generally yields biased results and is not recommended (see lines 69–71). Based on this feedback, we will better explain this point in the paper.}
>
> \textbf{W4: Thank you for this helpful feedback. We will add Thompson sampling as an additional baseline in our experiments (see lines 242-244), addressing this point directly.}
>
> While our experiments primarily compare PartiBandits to simple random sampling (SRS), we agree that it is valuable to consider additional baselines. In most practical settings, SRS and stratified random sampling (StRS) are the standard approaches—and often the only realistic choices available—since the effectiveness of more sophisticated methods is highly domain- and application-specific. In practice, whether alternative sampling algorithms outperform SRS or StRS depends crucially on the relationship between observed covariates and the outcome of interest, which may not always be known or exploitable. That said, we have discussed adaptive sampling methods, such as Thompson sampling and stratified adaptive cluster sampling, as possible additional baselines for future comparison. We will be conducting additional experiments involving Thompson sampling so that it can serve as an additional baseline for comparison. It should be appropriate for use with the AFC data and can also be naturally incorporated into the Monte Carlo experiments we have already conducted.
>
> \textbf{W5: Thank you for pointing out these presentation issues. We agree that the notation for $\rho_{\leq}$ and $\rho_{>}$ in Theorem 4 could be confusing. These symbols were not meant to denote special quantities in themselves, but rather to communicate that less than a quarter of labels were flipped on each side of the partition; we will revise the statement to clarify this intent and remove any ambiguity. Regarding $R_1$, it is not explicitly defined in the main text and was included only to help readers locate the corresponding quantity in Aznag et al. (2023) for comparison. As it is not essential for our main discussion, we will either clarify the reference or remove it to avoid confusion.}
>
> \subsection{Questions}
>
> \textbf{Q1: Thank you for raising this question. We address this in point W1.}
>
> \textbf{Q2: Thank you for this question. Please see also points W4 and Q3 in our response to Reviewer ehtb. The dependence on $\tau$ is as follows: when $\tau$ is small, we rely more heavily on the Variance-UCB adaptive sampling strategy, which is reasonable if the label budget is large and/or $\sigma_{\min}$ is not too small. When $\tau$ is large, the procedure defaults to simple stratified random sampling, ensuring the stated dependence on the label budget in the theorem but resulting in slightly less optimal constants. In the final bounds, $\tau$ essentially appears as a constant in the denominator, but for sufficiently large label budgets, its effect becomes negligible as the guarantees of the Variance-UCB procedure dominate. In response to this feedback, we will better explain this dependence and trade-off more explicitly in the revised text.}
>
> \textbf{Q3: Thank you for raising this question. We believe we have addressed this point in our response to W3 above.}

---

> > ### Author Response · Authors · 2025-08-06
> >
> > Thank you very much once again for your thoughtful and detailed review, and for your engagement throughout the process. Your feedback has been instrumental in helping us refine this work.
> >
> > Following the suggestion of the area chair, we are writing to inquire about whether all concerns raised have been addressed.  For reference, it may be helpful to review our discussion with Reviewer ZBCc, who raised several similar points; after considering our responses and revisions, Reviewer ZBCc indicated that their main concerns had been addressed. We are happy to make additional revisions or provide further clarification if needed. We would also like to kindly reiterate our note about the consideration of a possible score increase if the main concerns appear to have been addressed.
> >
> > Of course, we fully understand if you prefer to wait until the remaining reviews are finalized or until you have had additional time to reflect. Thank you again for your support in improving our work, and please let us know if you have any additional questions or comments.

---

> ### Comment · Reviewer_MZfC · 2025-08-08
>
> Thank you for the detailed rebuttal! The authors managed to make a significant addition to more general cases on the top of hard margin condition and binary setup, explain the role of \tau more explicitly, add the Thompson sampling baseline, and revise the confusing notations. This is impressive and I tend to raise my score to accept.

---

> > ### Author Response · Authors · 2025-08-08
> >
> > Thank you for your thoughtful response, and for your encouraging feedback on our rebuttal! We will incorporate these additions, clarifications, and revisions into the paper as planned, and we greatly appreciate your consideration of a score increase. Thank you once again for your engagement and support in improving our work.

---

### Official Review · Reviewer_Wv21 · 2025-06-27

**Clarity:** 4
**Significance:** 3
**Originality:** 3
**Rating:** 5
**Confidence:** 2

**Summary:**

This paper proposes a PartiBandits algorithm for estimating the mean of a binary random variable with auxiliary information, which includes two stages: (1) Using a Agnostic Active algorithm to learn a partition of the data to shrink the average conditional variance of the variable; (2) Using a UCB-style subroutine to actively request labels for the learned strata. The paper proves that the algorithm could achieve a near-exponential improvement in estimation accuracy in comparison to Simple Random Sampling, and the convergence rate is minimax optimal. This paper also provides empirical verification of the algorithm's performance using real-world data.

**Questions:**

1. Is it difficult to extend the algorithms and theoretical results to non-binary cases?

2. How is the computational cost of the PartiBandits? Could it be applied to large datasets in practice?

**Ethical Concerns:**

["NO or VERY MINOR ethics concerns only"]

**Limitations:**

The Bayes Optimal Classifier is in the Class assumption (Assumption 2) could be too strong for practical settings, despite it is common in theoretical analysis.

**Paper Formatting Concerns:**

No major concerns for paper formatting.

**Quality:**

4

**Strengths And Weaknesses:**

Strength: This paper shows strong results and proposes algorithms that achieve theoretically-optimal convergence rates. It also bridges a gap between UCB algorithms and disagreement-based methods. This paper provides both theoretical and empirical evidence to support the claims. It is also well-structured and the presentation of the main results are clear.

---

> ### Author Rebuttal · Authors · 2025-07-31
>
> \textbf{Q1}: \textbf{Thank you for raising this important question. Extending the near-exponential savings result to non-binary cases would greatly strengthen our contribution. To achieve this, we revised the main theorem and added Corollary 3, as detailed in Point 0 of our response to Reviewer ZBCc.}
>
> \textbf{Q2}: \textbf{Thank you for this question. Active learning algorithms like $A^2$ can be computationally intensive because of their sequential nature. To improve scalability, we revised the PartiBandits pseudocode and main theorem (see Point 0 in our response to Reviewer ZBCc) to support alternative, more computationally efficient active learning subroutines for classification. This flexibility allows us to readily incorporate faster algorithms (whether now or in the future) into the PartiBandits framework for large-scale applications.}

---

> > ### Comment · Reviewer_Wv21 · 2025-08-05
> >
> > Thank you for your response and clarification. I decided to keep my score.

---

> > > ### Author Response · Authors · 2025-08-06
> > >
> > > Thank you very much for your note and for your careful engagement with our work. Please let us know if you have any further questions or comments as the review process continues.

---

### Official Review · Reviewer_ehtb · 2025-07-02

**Clarity:** 2
**Significance:** 4
**Originality:** 3
**Rating:** 5
**Confidence:** 3

**Summary:**

The paper introduces a novel (meta) algorithm for the estimation of the mean of a binary random variable Y that depends on some (informative) X in a scenario where labels are expensive.  The proposed algorithm consists of two parts: first, an active-learning approach learns a partition G of the space X so that the partitions $A_1,\dots, A_G$ with probability $P_k=P(X\in A_k)$ minimize Var(Y|A_k). This is done using an online classifier that partitions based on its predictions. Then in a second step a bandit algorithm is applied which selects the partition A_k so that a bound on the weighted variance $P_k^2 Var(Y|A_k)$ is minimized. The first part of the algorithm is a direct application of the existing algorithm $A^2$, while the second part is a slight variation of Variance-UCB.The idea is that the initial split reduces the variance based on how informative X is, while the second part then focuses the majority of the remaining sample budget on the high variance partitions. The work establishes matching lower and upper bounds for each algorithm, where the lower bound is given by example. Finally, two small experiments illustrate the work.

**Questions:**

1. Can you provide experimental results with more difficult settings, e..g, harder to separate classes?
2. Can you provide scaling results for Algorithm 2 as g increases?
3. Can you sopecify what the significance of tau is in Algorithm 1, and how your results change when tau is set to 0?

**Ethical Concerns:**

["NO or VERY MINOR ethics concerns only"]

**Final Justification:**

The authors answered all points I raised, and also the ones raised by other reviewers. Additional experiments were added and the discussion did not uncover any fundamental flaws in the maths or experiments. To the contrary, the authors expanded their analysis to cover the issues raised in form of additional corollaries and thus greatly reduced my feeling of the work being "minimalistic". I think this strengthens the paper significantly.

While I am not an expert in the area, I see that the other reviewers express happiness with the response, and indicate a possible raising of their scores. This was the best paper in my batch, and in light of all the above, I feel safe to propose acceptance of this paper at NeurIPS.

Finally, I would like to thank the authors for a very honest rebuttal and discussion, with clear indication of weaknesses and mistakes.

**Limitations:**

no. A conclusion and discussion is missing. I mentioned above the missing discussion of the scaling behavior in several cases.

**Quality:**

3

**Strengths And Weaknesses:**

I was not able to conduct a proper review since the work is very far outside my core domain and i only realized very late that the total review load is around 150 pages and since I scheduled this paper last due to its complexity, I was not able to read up on related work. I still do a best-effort review based on the main article without appendix. I am choosing low confidence and opt for weak accept.

The strength of the paper is a clear theoretical analysis of the problem including matching lower and upper bounds. This is based on the reliance on strong results from prior work, but is still an achievement.

The main weaknesses i would like to point out is the overall minimalistic approach of the paper
1. Algorithm 2 seems underexplored and minimalistic.
2. Not all sources of uncertainty seem to be carried over to the analysis
3. the experimental part is very short and especially experiment 2 is not understandable to me. The experiment do not provider enough information to establish the theoretical rates and are too minimalistic to properly explore strengths and weaknesses.
4. In general, bounds and their intuitive meaning are underexplored.

Details:
1. From my understanding, Algorithm 2 partitions the input space in just two parts: A_0 and A_1 based on the classification of the hypothesis produced by the A2 algorithm. As a result, the gain of Algorithm 2 is just based on how well the classifier predicts the right class but does not make use of variability of regions within a class. E.g., assuming the true data is generated by logistic regression, we would ideally want to partition in three classes, where the middle class is close to the boundary.
2. Algorithm 1 assumes that $P_k=P(x\in A_k)$ is known. This either means that the assumption is that the initial unlabeled dataset is the full distribution, or the values are given by an oracle. I do not think that matters for small number of partitions since we can assume that the amount of unlabeled data is always sufficiently large compared to the amount of labeled data that the additional error can be included in some of the factors, but formally it is not discussed in the theorems.
3. Experiment 2: Do I understand it correctly, that in each trial from the 10000 drawn people only the ones are selected which fall in the 5% lower and upper intervals, that is we see the results based on easily separable classes? In that case, I think the experiment is almost a "strawman" ( for a lack of better word) since in this example it is also incredibly easy to find the classifier that is separating them with just a few samples. In this case, a maximum margin classifier with just 1 sample from each class would have a high probability to be bayes optimal (when both labels are correct). So this experiment shows the extreme case of where the algorithm must perform best. I still do not understand the shown figure, because it appears the result is almost constant, even though i would expect to see 1/N convergence in both algorithms. A larger sample range, especially in the small sample regime would be interesting. Similarly, for experiment 1 only the case of symmetric noise and two same size classes is investigated. I do not find an explanation in the paper why this special case should be sufficient, since in this case both partitions should have identical variance.
4. I would have loved to read more about the intuitive meaning of the bounds. For Algorithm 2 for example, I do not understand why it should not depend on the number of partitions. I would intuitively expect the bound to become worse as the number of partitions grows. Similarly, for Algorithm 1 i do not understand why the warmstart is necessary, since it does not appear in the bound.


technicalities:
1. I was not able to find the warmstart component in Algorithm 1. I think the initialisation of the std-dev in line 1 should do that, but then i would expect some dependency of t on g(not G) to ensure that each g has an area where its variance is maximized.
2. A number of quantities appear to be undefined in the main article. For me as a non-expert, especially considering the density or the article, it made me search quite a few times because i was not sure whether i missed something. It might be that some are standard notation, but i would like the authors to check the following variables that i mark with the first occurrence i found:
- $\tilde{\mathcal{O}}$ line 65 (is this: $\mathcal{O}$ ignoring log-terms?
- $R_1$, line 201
- $c_1,c_2$ line167.
- $ \Sigma_1(\mathcal{G})$ line 178
3. Theorems do not clearly state assumptions. I know that you stated assumptions globally, but I think it is crucial to describe concrete assumptions of each analysis as minimal as possible, e.g., Theorem 3 should not depend on Assumptions 1 and 2 since the algorithm assumes $\mathcal{G}$ is given. At the same time, considering the choice of $c_1,c_2$ in Algorithm 2, it appears that Theorem 3 holds for a much broader set of random variables than binary Y.

---

> ### Author Rebuttal · Authors · 2025-07-31
>
> Thank you very much for your thoughtful review. Your feedback raises a number of valuable points and questions, and these have helped shape our plans to revise the paper with important clarifications and additions. Below, we provide responses to each point, with the \textbf{main responses in bold}. After reviewing the clarifications and planned revisions below, we hope there will be consideration of whether a re-evaluation of the score is warranted.
>
> \subsection{Strengths, Weaknesses, and Technicalities}
>
> \textbf{W1}: \textbf{Thank you for highlighting this point. Your feedback led us to write Corollary 2 (see Point 0 in our response to Reviewer ZBCc), which shows how to construct partitions with more than two strata and leverage additional structure in the covariate space.}
>
> Corollary 2 covers cases where we can leverage multiple strata, even with binary random variables. Specifically, Algorithm 4.2 of Puchkin and Zhivotovskiy (2022), which forms the basis for Corollary 2, combines standard binary classification with an abstention mechanism. This approach lets the analyst partition the covariate space into three regions: where the learned classifier predicts 1, where it predicts 0, and where it abstains (predicts $(*)$). These correspond to three strata: likely 1’s, likely 0’s, and ambiguous cases (see Point 0 in our response to Reviewer ZBCc). Our framework and Corollary 2 therefore support the kind of multi-strata setup you describe, even in the binary case.
>
> The main guarantee of exponential savings (i.e., the rate in Theorem 5) remains intact, and this approach enables the algorithm to leverage three strata as you suggest. While further adjustments to the main results from Puchkin and Zhivotovskiy may be needed to optimize the final rate, experiments based on this framework are likely to exhibit improved performance, as you note. We will include additional details in both the main text and Appendix to address your feedback on the number of strata and elaborate on these points.
>
> \textbf{W2}: \textbf{Thank you for raising this point about the assumption that the $P_k$'s are known. Our analysis assumes we know the $P_k$'s and that we have enough unlabeled data to estimate these probabilities accurately. This assumption is standard in the active learning literature (see, for example, Puchkin and Zhivotovskiy (2022) and Hanneke (2011)), which typically presumes access to a large pool of unlabeled data. Your observations about the impact of few versus many partitions are correct. We have revised the paper to clarify this point in response to your feedback.}
>
>
> \textbf{W3}: \textbf{Thank you for your thoughtful comments on Experiment 2 and the experimental design. You are right that focusing only on the most easily separable classes would create an overly favorable scenario, allowing a classifier to achieve near-perfect separation with few samples. However, Experiment 2 takes a different approach: our outcome is the interaction variable $HB$, not just a class label, which adds complexity. For example, even if a patient is very likely to be Black ($B=1$) based on covariates, they may not have hypertension ($H=1$), so $HB$ is not always $1$ in the upper tail. Because the AFC data are not IID draws from the general population, the upper tail may, for example, include individuals from predominantly Black but affluent neighborhoods who are less likely to have hypertension. Thus, $HB$ may even be $0$ more often than for those classified as $B=0$, so the class separation is not immediate. It would be, however, if the outcome of interest was just $B$, since the covariate represents $\Pr(B = 1)$ so it would be precisely the upper and lower part of the distribution of the outcome, unlike the case when the outcome is $HB$ instead. This setup reflects the reality of many datasets that are not IID samples from the general population. We will clarify this distinction in the paper in response to your feedback.}
>
> Additionally, your observations about the figure prompted us to review our code. We found that we had applied the quantile operation in reverse—calculating lower rather than upper quantiles—and had not correctly implemented random sampling for SRS. After correcting these issues, we reran Experiment 2 with the proper sampling and a larger label budget range. The updated figures now reflect the trends you suggested. For Experiment 1, we also ran additional experiments with different noise levels, data generating processes (logit and probit), asymmetric class distributions, and degrees of class separation. Across these expanded results, PartiBandits continues to show consistent performance. We are ready to incorporate these findings into the revised paper.
>
> \textbf{W4}: \textbf{Thank you for your comments on the intuitive meaning of the bounds, especially regarding the number of partitions in Algorithm 2 and the role of the warmstart parameter in Algorithm 1. We agree these points are important and have added further clarification to the text.}
>
> You are correct that the performance of stratified sampling algorithms generally depends on the number of strata. We have updated the PartiBandits pseudocode and main theorem to explicitly accommodate more strata and to show near-exponential rates (see Point 0 in our response to Reviewer ZBCc). For the revised theorem statement, we will clarify the dependence on the number of strata in the main text.
>
> The warmstart in Algorithm 1 serves as a "buffer," ensuring that Variance-UCB remains effective when the overall label budget is small and the algorithm might otherwise undersample low-variance groups. As Aznag et al. (2023) discuss, the regret of Variance-UCB can scale inversely with $\sigma_{\min}$, the smallest group variance. By assigning each group a minimum number of samples, the warmstart helps guarantee the algorithm’s fast-rate properties. For small $N$, the risk bound behaves as $\widetilde{\mathcal{O}}( 1/\tau N)$, where $\tau$ is the warmstart proportion. As $N$ grows and the fast-rate conditions hold, the dependence on $\tau$ vanishes and the sharper rates of Aznag et al. apply. We followed Aznag et al. in focusing only on the dependence on the label budget in the bound, but we will clarify this dependence and its implications in the text of the Theorem.
>
> \textbf{T1}: \textbf{Thank you for your careful reading of Algorithm 1 and for highlighting the role of the warmstart component. You are correct that the initialization of the standard deviation in line 1 serves as the warmstart. Our procedure does not include an explicit dependence of $t$ on $g$; instead, the warmstart ensures that when the label budget is too small for Variance-UCB’s rate guarantees, we fall back to uniform (StRS) sampling (see Point W4 above). We have revised the text to clarify this point.}
>
> \textbf{T2: Thank you for drawing attention to these points regarding notation. Most of the notation is standard, but we have clarified some of it in the revisd text.}
>
> 1. Yes, $\widetilde{\mathcal{O}}$ ignores logarithmic factors and fixed constants.
> 2. We did not explicitly define $R_1$ in the main text; we referenced it only to help readers compare with Aznag et al. (2023). Since $R_1$ is not essential to our main discussion, we will either clarify its reference or remove it to prevent confusion.
> 3. $c_1$ and $c_2$ are simply the subgaussianity constants (see Aznag et. al (2023)), but we will clarify them in the revised paper.
> 4. $\Sigma_1(\mathcal{G})$ is described in English in lines 72–73 (as the average within-group variance of the outcome given stratification $\mathcal{G}$), and is defined formally in the Appendix. We agree it would be helpful to include the formal definition directly in the main text, and we will do so.
>
> \textbf{T3}: \textbf{Thank you for raising this important point about the clarity of assumptions in our theorems. You are correct that Theorem 3 does not rely on Assumptions 1 and 2 and holds for a much broader class of random variables than just binary $Y$. We will revise the paper to state the assumptions for each theorem explicitly and clarify that Theorem 3 applies in this greater generality.}
>
> \subsection{Questions}
>
> \textbf{Q1: Please see our response to point W3 above.}
>
> \textbf{Q2: Thank you for this question about how Algorithm 2 scales as the number of groups $g$ increases. We have addressed the theory behind this question in Corollary 3 (see Point 0 in our response to Reviewer ZBCc).}
>
> \textbf{Q3}: \textbf{Thank you for highlighting the significance of $\tau$ in Algorithm 1. The parameter $\tau$ safeguards the Variance-UCB procedure by ensuring that part of the label budget is allocated to simple stratified random sampling (StRS), which maintains theoretical guarantees even when the label budget or $\sigma_{\min}$ (the smallest within-group variance) is small. Setting $\tau = 0$ allocates all labels via Variance-UCB, achieving optimal rates when both $\sigma_{\min}$ and the label budget are sufficiently large. However, a positive $\tau$ ensures we sample from every group, preserving the guarantee of Theorem 3 with only slightly less optimal constants. For large label budgets, the effect of $\tau$ disappears and the optimal rates of Aznag et al. (2023) apply. Since the rate’s dependence on the label budget remains unchanged (only the constants differ), we did not state this dependence explicitly in the theorems. We will clarify these points and add more detail to the theorems in response to your question.}

---

> > ### Comment · Reviewer_ehtb · 2025-08-03
> >
> > I thank the authors for their detailed reply. I am happy to read that you could find a mistake in your plots, then this review still served a purpose.
> >
> > I am still keeping the weak accept for now since I am not knowledgeable enough of the domain to argue whether this is a high enough impact contribution and I will follow the recommendation of the other reviewers who are on the edge in case they change their grade.

---

> > > ### Author Response · Authors · 2025-08-06
> > >
> > > Thank you very much for your follow-up and for your careful engagement with our work. We appreciate your thoughtful feedback, which helped us significantly improve the paper. Please let us know of any additional questions or comments you may have as the review process continues.

---

### Official Review · Reviewer_ZBCc · 2025-07-03

**Clarity:** 3
**Significance:** 3
**Originality:** 2
**Rating:** 4
**Confidence:** 3

**Summary:**

The authors consider the problem of estimating the mean of a binary random variable $Y$ using limited number of labelled instances. They assume that they have an access to covariate $X$ that may be informative about $Y$. The authors suggest a PartiBandits algorithm, which takes a hypothesis class $\mathcal C$, label budget $N$, a confidence level $\delta$, and a buffer fraction $\tau$ as an input. The algorithm performs in two steps. First, using the $A^2$ (agnostic active) algorithm as a subroutine, PartiBandits stratifies the covariate space into two sets. After that, it estimates stratified mean using Variance-UCB algorithm [Aznag et al., 2023] with a ''warm-start'' step (WarmStart-UCB). Assuming the Tsybakov noise condition and that the Bayes optimal classifier belongs to the concept class $\mathcal C$, the authors prove minimax optimal upper and lower bounds on the accuracy of mean estimation. They illustrate advantage of their procedure over stratified random sampling using Health Record Data.

**Questions:**

1. How the result of Theorem 5 will change if $Y$ is a (non-Bernoulli) random variable taking its values in $[0, 1]$?

2. Is it possible to further improve the algorithm by increasing the number of strata?

3. In [Puchkin and Zhivotovskiy, 2022], the authors show that exponential improvements in agnostic active learning are possible even if the Bayes optimal classifier does not belong to the concept class $\mathcal C$. Instead, they require $\mathcal C$ to have a bounded combinatorial diameter and a finite star number. Is it possible to replace Assumption 2 by similar requirements in your setup?

4. Is stratified random sampling the only existing competitor? Can you compare PartiBandits with other algorithms?

**Ethical Concerns:**

["NO or VERY MINOR ethics concerns only"]

**Final Justification:**

The authors have thoroughly addressed my main concerns, namely Weaknesses 1-3 in my initial review. The authors extended the setup for multiclass and multistrata scenarios (see Corollary 2 in their rebuttal). Besides, relying on findings of Puchkin and Zhivotovskiy (IEEE Trans. Inf. Theory, 2022), they showed (Corollary 3) that nearly exponential savings are still possible even if the Bayes optimal classifier is not in the concept class.

**Limitations:**

Yes.

**Paper Formatting Concerns:**

No.

**Quality:**

3

**Strengths And Weaknesses:**

Strenghts.

1. According to the numerical experiments, the PartiBandits algorithm significantly outperforms stratified random sampling.

2. The authors prove rigorous guarantees on the PartiBandits performance. They also prove a matching lower bound claiming optimality of their procedure.


Weaknesses.

1. The setting is limited to binary outcomes.

2. The assumption that the Bayes classifier must belong to the concept class $\mathcal C$ (Assumption 2) is rarely satisfied in practice.

3. The number of strata is limited to 2.

4. The proofs of the upper and lower bounds (Theorems 5 and 6) significantly rely on the well-known results from the literature on active learning (for example, Theorem 4 from [Hanneke, 2011]).

---

> ### Author Rebuttal · Authors · 2025-07-31
>
> We thank you very much for your careful review. It raises many good points and questions that have informed key revisions we have made to the paper. In what follows, we provide responses to each of the issues, with the \textbf{main points in bold}. However, there is one point we would like to discuss first as it addresses many of the issues raised. Once the clarifications and plans for revision below have been reviewed, we kindly ask the reviewer to consider if a re-evaluation of the score may be possible.
>
> \textbf{Point 0: In response to feedback about the paper’s restrictive assumptions (e.g., Tsybakov noise, Bayes classifier in class) and binary problem setup, we revised Step 2 of the PartiBandits algorithm and revised our main result of near exponential savings in Theorem 5. These limitations arose simply because we used the classical $A^2$ algorithm in Step 2 for clarity and intuition. As noted in lines 234–245, one can substitute less restrictive subroutines as needed. To make this explicit, we updated the PartiBandits pseudocode and reformulated the main theorem to accommodate broader assumptions and setups. Only minor adjustments to the proof are required, as detailed below.}
>
>
> \textbf{The revised theorem allows us to incorporate any algorithm for classification into the PartiBandits framework through Step 2, provided it achieves exponential savings. PartiBandits directly inherits the assumptions and problem setups of any such algorithm, ensuring it remains effective for efficient mean estimation as active learning methods continue to advance.}
>
>
> \textit{What we changed.} In Step 2 of PartiBandits, we now allow any algorithm, $\mathcal{S}$, that learns a binary or multiclass classifier $\widehat{f}$ satisfying, with high probability, $\mathbb{E}[(\widehat{f} - y)^2] \lesssim \nu + \exp\left(c \cdot \left(-\frac{N}{\log(N)}\right)\right)$, where $\nu$ is the infimum risk over the hypothesis class, $c$ is a constant, and $N$ is the label budget ("Condition 1"). With this modification, along with minor updates to the rest of the PartiBandits pseudocode, we reformulated the main theorem to allow more flexibility in assumptions and problem setups. The revised theorem is as follows.
>
> $$***$$
>
> \textbf{Theorem 5}: Suppose that for some joint distribution of $(X,Y)$ where $Y$ is a $k$-class random variable, there is a subroutine, $\mathcal{S}$, that learns a $k$-valued classifier $\widehat{f}$ such that with high probability, $\mathbb{E}[(\widehat{f} - y)^2] \lesssim \nu + \exp\left(c \cdot \left(-\frac{N}{\log(N)}\right)\right)$ where $\nu$ is the infimum risk of all possible classifiers in the hypothesis class, $c$ is a constant and $N$ is the label budget. Then, if $\mathcal{S}$ is used as the subroutine in Step 2 of PartiBandits (Algoirthm 2), we have:
>
>     $$
>     \left| \widehat{\mu}_{\text{PB}} - \mathbb{E}[Y] \right|^2 = \tilde{\mathcal{O}}\left( \frac{\nu + \exp(c \cdot (-N/\log(N))) }{N} \right),
>     $$
>
>     where $c >0$ is a constant.
>
> $$***$$
>
> \textit{Why this change is possible.} We can make this change because the only requirement for Step 2 is to identify strata that are likely to be homogeneous with respect to the outcome, a task that any well-performing classification algorithm can achieve. In the original version of PartiBandits, we used the $A^2$ subroutine in Step 2 to identify homogeneous regions of the covariate space, but any suitable classification algorithm can fill this role. Intuitively, a classifier with low excess risk tends to group together samples with similar labels, which helps us define more homogeneous strata. This structure enables us to obtain stable mean estimates within each stratum and efficiently aggregate them to efficiently estimate the overall population mean.
>
> We can formalize this argument as follows. For binary $y$, Theorem 5’s rate holds as long as the classifier used in Step 2 of PartiBandits satisfies Condition 1 above. We show in the proof of Theorem 5 (see Appendix) that the average within-stratum variance is bounded above by $\mathbb{E}[(f(x) - y)^2]$ (where we change notation so that $f$ in this paragraph is now the learned classifier). To see this, we first apply the law of total expectation to write $\mathbb{E}[(f - y)^2] = \mathbb{E}[(f - y)^2 \mid f = 1] P(f = 1) + \mathbb{E}[(f - y)^2 \mid f = 0] P(f = 0)$. Applying the bias-variance decomposition, we obtain $\mathbb{E}[(f(X)-Y)^2] \geq \Var(Y \mid f=1) \Pr(f = 1) + \Var(Y \mid f=0) \Pr(f = 0)$. This right-hand side is exactly the average within-group variance that determines the high-probability bound on the estimation error for the PartiBandits population mean. This derivation is purely algebraic and does not rely on any assumptions specific to the active learning algorithm used in Step 2. The same argument extends directly to multiclass $y$. We now outline a few corollaries that illustrate how PartiBandits adapts to different assumptions and setups through the choice of Step 2 subroutine.
>
> Corollary 1: Suppose $Y$ is binary and the Bayes optimal classifier is in the class and the hard margin condition is satisfied. Then the rate of Theorem 5 holds with $\mathcal{S}$ being $A^2$ (this was the original result of our main Theorem).
>
> Corollary 2:  Suppose $Y$ is binary and the hypothesis class has bounded combinatorial diameter and a finite star number. Then the rate of Theorem 5 holds with $\mathcal{S}$ being Algorithm 4.2 of Puchkin and Zhivotovskiy (2022).
>
> Corollary 3: Suppose $Y$ is a $k$-class random variable and the covariate space is such that the conditions necessary for Corollary 2 in Agarwal (2013) hold. Then, an analog of the near exponential savings rate of Theorem 5 holds with $\mathcal{S}$ being Algorithm 1 of Agarwal (2013).
>
> \subsection{Strengths and Weaknesses}
>
> \textbf{W1:} \textbf{We have added Corollary 3 to address this point (see Point 0).}
>
> \textbf{W2}: \textbf{We have added Corollary 2 to address this point.}
>
> As you correctly note in Question 3, the algorithm of Puchkin and Zhivotovskiy (2022) achieves exponential savings under weaker assumptions. As we explain in Point 0, using this algorithm as the subroutine $\mathcal{S}$ in Step 2 of PartiBandits allows PartiBandits to inherit these more flexible assumptions, so an analogous near-exponential savings result follows.
>
> \textbf{W3: We have added Corollary 2 to address this point.}.
>
> We discuss this point more in point W1 in our response to Reviwer ehtb.
>
> \textbf{W4}: \textbf{It is true that our proofs rely on existing active learning results, but this is a feature, not a bug. We designed the proof of Theorem 5 to apply to any active learning classification algorithm used in Step 2 of PartiBandits, as long as it reduces excess risk with exponential savings (Condition 1). This is thanks to an original decomposition we derive that bridges classification tasks and mean estimation tasks, as explained above. As active learning is a rapidly developing field with more advanced algorithms being developed every year, we wanted to ensure all such algorithms for classification, either now or in the future, can be incorporated into the PartiBandits framework (by way of Step 2) to help with efficient mean estimation. In response to this feedback, we have revised the Pseudocode of PartiBandits and the statement of our main Theorem proving the near exponential rates to make this contribution clearer.}
>
> \subsection{Questions}
>
> \textbf{Q1}: \textbf{Thank you for raising this question. We can extend the PartiBandits framework to general real-valued outcomes by discretizing $Y$ and applying Corollary 3, which addresses the feedback about the non-binary case. Achieving similar results without discretization is substantially more difficult and would likely require new advances in active learning for "classifying" continuous outcomes. To our knowledge, no one has yet shown exponential savings in the continuous outcome setup without discretization, for classification, estimation, or otherwise. Success on that front would itself merit a separate paper. We will revise the manuscript to explain these points.}
>
> The idea behind the discreitzation approach is to partition the range of $Y$ into $k$ intervals and treat each as a separate class; for example, mapping $Y \in [0,1]$ into $k=10$ bins such as $[0,0.1), [0.1,0.2), \ldots, [0.9,1]$.
>
> \textbf{Q2}: \textbf{Thank you for raising this question. Corollary 2 (see Point 0) has been written to directly address this, demonstrating that such an improvement is indeed possible.}
>
> Please see our response to point W3 and point W1 in our response to Reviwer ehtb for further details.
>
> \textbf{Q3}: \textbf{Thank you for raising this question. This understanding of the work of Puchkin and Zhivotovskiy (2022) is correct, and yes, Assumption 2 may be replaced by similar requirements in their setup. We have written Corollary 2 to address this feedback (see Point 0).}
>
> See also our response to point W2 above.
>
> \textbf{Q4}: \textbf{Thank you for this helpful suggestion—we will add Thompson sampling as an additional baseline in our experiments (see lines 242-244), addressing this point directly.}
>
> We address this in section W4 in our response to Reviewer MZfC.

---

> > ### Comment · Reviewer_ZBCc · 2025-08-02
> >
> > Thank you very much for the detailed answers. I hope that the corresponding changes will be incorporated into the final version of the paper.

---

> > ### Comment · Reviewer_ZBCc · 2025-08-05
> >
> > Following the suggestion of the area chair, I would like to indicate that I am satisfied with authors' response. Corollaries 2 and 3 address my main concerns. I have no further questions.

---

> > > ### Author Response · Authors · 2025-08-06
> > >
> > > Thank you very much for your confirmation and for your thoughtful feedback. Your comments and suggestions were indispensable in helping us strengthen the paper. We would like to kindly reiterate our note about the consideration of a possible score increase, if this has not already been implemented, now that your main concerns have been addressed. Of course, we fully understand if you prefer to wait until the remaining reviews are finalized or until you have had additional time to reflect. Thank you again for your engagement and support in improving our work, and please let us know of any additional questions or comments you may have.

---

> > > > ### Comment · Reviewer_ZBCc · 2025-08-07
> > > >
> > > > I will finalize my score after discussion with other reviewers and the area chair. But I believe that you made a good job during the rebuttal and that Corollaries 2 and 3 will be incorporated into a revised version of the paper. For this reason, I am tending to increase my score.

---

> > > > > ### Author Response · Authors · 2025-08-07
> > > > >
> > > > > Thank you for your response, and for your positive feedback on the rebuttal. We are indeed planning to incorporate these revisions into the paper. We deeply appreciate your inclination toward a score increase. Please do let us know if you have any further questions or comments in the meantime.

---

> > > > > > ### Comment · Reviewer_ZBCc · 2025-08-07
> > > > > >
> > > > > > I have no further questions.

---

> > > > > > > ### Author Response · Authors · 2025-08-08
> > > > > > >
> > > > > > > Thank you for your confirmation. We look forward to your final determination.

---

### Comment · Area_Chair_Qbq2 · 2025-08-05
**Reminder to Reviewers to engage in discussion with Authors**

Dear Reviewers,

Please read the authors' rebuttals if you have done so yet, and respond to them as soon as possible to allow sufficient time for follow-up exchanges. The Author-Reviewer discussion is crucial to a constructive reviewing process, to which your reactivity and engagement are indispensable.

Best regards,

AC

---

### Note · Authors · 2025-08-15

We are grateful to all reviewers for their thoughtful, constructive, and engaged feedback. Their comments helped us generalize our main result to cover settings with non-binary random variables, cases where the hard-margin or Bayes-classifier-in-class assumptions do not hold, and cases where more than two strata are desirable. They also prompted us to clarify notation, assumptions, intuitions, and the roles of certain parameters, and to refine our experimental setups and add new baselines. We appreciate that the reviewers indicated that these changes will strengthen the paper and expressed an inclination to raise their scores.

The planned revisions to the PartiBandits algorithm and to the main theorem on near-exponential savings are simple to implement. As already noted in the paper (lines 224–235), an alternative subroutine to $A^2$ may be used in Step 2. All that is required to extend our result to more general problem setups, such as those mentioned above, is to make explicit how to substitute different algorithms into Step 2 of PartiBandits. As shown in the discussion with Reviewer ZBCc (“Point 0”), the current proof of the near-exponential savings result is easily amenable to this change.

We also strengthened the experimental evaluation in direct response to reviewer suggestions. We ran experiments with varied noise levels, more data-generating processes (logistic and probit), asymmetric class distributions, and differing class separations. Near-exponential savings are still observed in these new experimental setups. We are also adding Thompson sampling as an additional baseline. We expect PartiBandits to outperform this other baseline, since the asymptotic rate of adaptive Thompson sampling is of the same order in the label budget as SRS [1], whereas PartiBandits has a provably more optimal dependence on the label budget (near exponential).

[1] Félix-Medina, Martín H. "Asymptotics in adaptive cluster sampling." Environmental and Ecological Statistics 10.1 (2003): 61-82.

---

### Decision · Program_Chairs · 2025-09-17

**Decision:**

Accept (poster)

**Comment:**

This article explores the disagreement-based active learning framework to improve the sample efficiency for the population mean estimation of a target variable $Y$, enabled by the information of covariates $X$ which are predictive of $Y$. In the proposed algorithm "PartiBandits", a predictor of $Y$ is first obtained by a disagreement-based active learning method, allowing the construction of homogenous subgroups w.r.t. to the value of $Y$. Then the population mean estimation is carried out from these subgroups with a UCB (upper confidence bounds)-style procedure. The minimax optimality of "PartiBandits" is demonstrated with matching upper and lower bounds. It is also tested on a real-world data set against the standard baseline of random sampling.

The main issues raised by Reviewers concern the strong assumptions for the theoretical results and the restrictive setting for the deployment of the algorithm. The authors effectively addressed these limitations, many of which are simply removed by using more advanced alternatives for disagreement-based active learning. Reviewers were satisfied with the rebuttal, and recognized the interest and the theoretical strength of the proposed method.